# Snf1 AMPK positively regulates ER-phagy via expression control of Atg39 autophagy receptor in yeast ER stress response

Tomoaki Mizuno[ID]*, Kei Muroi[ID]¤, Kenji Irie[ID]

Department of Molecular Cell Biology, Faculty of Medicine, University of Tsukuba, Tsukuba, Japan

¤ Current address: Occupational and Aerospace Psychiatry Group, Graduate School of Comprehensive Human Sciences, University of Tsukuba, Tsukuba, Japan
* mizuno@md.tsukuba.ac.jp

## Abstract

Autophagy is a fundamental process responsible for degradation and recycling of intracellular contents. In the budding yeast, non-selective macroautophagy and microautophagy of the endoplasmic reticulum (ER) are caused by ER stress, the circumstance where aberrant proteins accumulate in the ER. The more recent study showed that protein aggregation in the ER initiates ER-selective macroautophagy, referred to as ER-phagy; however, the mechanisms by which ER stress induces ER-phagy have not been fully elucidated. Here, we show that the expression levels of *ATG39*, encoding an autophagy receptor specific for ER-phagy, are significantly increased under ER-stressed conditions. *ATG39* upregulation in ER stress response is mediated by activation of its promoter, which is positively regulated by Snf1 AMP-activated protein kinase (AMPK) and negatively by Mig1 and Mig2 transcriptional repressors. In response to ER stress, Snf1 promotes nuclear export of Mig1 and Mig2. Our results suggest that during ER stress response, Snf1 mediates activation of the *ATG39* promoter and consequently facilitates ER-phagy by negatively regulating Mig1 and Mig2.

## Author summary

All organisms are exposed to harmful environmental factors, for example, ultraviolet light, heat, and chemical compounds. These factors produce defective proteins within the cells both directly and indirectly by inducing genetic mutations. The endoplasmic reticulum (ER) is one of the cellular compartments and responsible for quality control of secretory and membrane proteins. The condition where defective secretory and membrane proteins accumulate in the ER is called ER stress. In human, ER stress is implicated in a variety of diseases, such as diabetes, cancers and neuro-degenerative diseases, including Alzheimer's disease and Parkinson's disease. To clear defective proteins in the ER, cell actuates the defense mechanism referred to as ER stress response. ER stress response is evolutionarily conserved among eukaryotic cells from yeast to human. Here, we investigate the mechanism by which ER stress induces ER-phagy, selective autophagic

**Funding:** TM received grant JP19K06632 from the Japan Society for the Promotion of Science KAKENHI (JSPS) https://www.jsps.go.jp/j-grantsinaid/. The funders had no role in study design, data collection and analysis, decision to publish, or preparation of the manuscript.

**Competing interests:** The authors have declared that no competing interests exist.

degradation of the ER, using the budding yeast as a model cell. We demonstrate the molecular link between the stress-responsive kinase, the transcriptional factors, gene expression of the autophagy-related gene and ER stress-induced ER-phagy.

## Introduction

The endoplasmic reticulum (ER) is an organelle responsible for the folding and modification of newly synthesized secretory and transmembrane proteins. Perturbation of ER homeostasis resulted from the changes in the intracellular and extracellular environments accumulates aberrant proteins in the ER lumen and membrane. This condition, which is called ER stress, has detrimental effects on whole cell homeostasis beyond the ER. Therefore, when ER stress is sensed, cells actuate adaptive responses to alleviate ER stress. In the budding yeast *Saccharomyces cerevisiae*, the unfolded protein response (UPR) signaling pathway composed of Ire1 and Hac1 plays a central role in ER stress response [1, 2]. Activation of the UPR in response to ER stress induces expression of genes encoding ER-resident chaperones and proteins functioning in ER-associated degradation (ERAD) [1, 2]. Besides, the UPR positively regulates expression of genes involved in lipid biogenesis and thereby increases ER size [3]. Thus, the UPR is undoubtedly essential for yeast cells to alleviate ER stress. Previous studies also demonstrated that the budding yeast ER stress response involves the signaling pathways including the stress responsive MAP kinases (MAPKs), such as Mpk1 and Hog1, and the Snf1 AMP-activated protein kinase (AMPK) [4–12]. They are activated by ER stress, modulate gene expression patterns, and regulate ER stress tolerance. Furthermore, it has been reported that ER stress elicits autophagy, an evolutionarily conserved process that mediates degradation and recycling of intracellular components [13, 14].

Autophagy can be categorized into two classes, microautophagy and macroautophagy [15, 16]. Macroautophagy generates double-membrane vesicles designated autophagosomes to transport intracellular components into the degradative compartment (the vacuole in the budding yeast). By contrast, cargo transport in microautophagy is achieved without utilizing autophagosomes. Macroautophagy can be further divided into two types that cause degradation non-selectively or selectively [15, 16]. Selective macroautophagy includes mitophagy, pexophagy, and ER-phagy, which specifically degrade mitochondria, peroxisomes, and the ER, respectively. Each selective macroautophagy requires the cargo-specific autophagy receptor, which localizes to a target for degradation and functions in the recruitment of the core autophagy-related proteins mediating the autophagosome formation. In the budding yeast, Atg32 and Atg36 act as an autophagy receptor for mitophagy and pexophagy, respectively [17–19]. Two autophagy receptors specific for ER-phagy, Atg39 and Atg40, have been identified in the budding yeast [20].

It has been demonstrated that non-selective macroautophagy is induced by ER stress in the budding yeast [13]. Additionally, there are reports showing that during yeast ER stress response, the ER is degraded by microautophagy [14, 21]. Furthermore, a recent study in which ER stress is induced by overexpression of the aggregation-prone secretory protein provided the evidence that ER stress triggers ER-phagy [22]. However, the mechanisms by which ER stress induces ER-phagy have not been fully elucidated. We therefore addressed this issue and found that autophagic degradation of the ER induced by ER stress is mainly dependent on Atg39. We also revealed that ER stress significantly increases *ATG39* promoter activity and this upregulation is diminished in cells deleted for the *SNF1* gene. Furthermore, ER stress-induced ER-phagy was effectively inhibited by *snf1* mutation. These *snf1* mutant phenotypes

were clearly suppressed by loss of Mig1 and Mig2 transcriptional repressors. Mig1 and Mig2 accumulate in the nucleus under unstressed conditions and their nuclear localization is reduced by Snf1 activation in response to ER stress. Taken together, these results suggest that during ER stress response, Snf1 negatively regulates Mig1 and Mig2, consequently promoting Atg39 expression and ER-phagy.

## Results

### ER stress induces ER-phagy

To investigate the mechanism by which ER stress induces ER-phagy, we used the strain expressing the carboxyl-terminally GFP-tagged Sec63 ER transmembrane protein. When ER-phagy is induced, Sec63-GFP is transported into the vacuole for degradation. Since GFP is resistant to the vacuole-resident proteases, autophagic degradation of Sec63-GFP yields free GFP [20]. Therefore, we examined the abundance of free GFP in unstressed and ER-stressed cells by western blot analysis using anti-GFP antibodies (Fig 1A and S1A Fig). Free GFP was undetectable under unstressed conditions. We could observe free GFP in cells treated for long periods (18 hours) with tunicamycin, which causes ER stress by inhibiting N-linked glycosylation, although free GFP was hardly detected in cells with short-term exposure to tunicamycin (~6 hours). We also observed free GFP production from Sec63-GFP using dithiothreitol (DTT), which causes ER stress by inhibiting the disulfide bond formation, as an ER stressor (S1B Fig). To confirm that free GFP production in response to ER stress results from autophagic degradation, we tested the effect of *atg1* mutation, which causes a defect in macroautophagy [15, 16]. In *atg1* mutant cells, free GFP was significantly decreased compared with wild-type cells (Fig 1A and S1B Fig), indicating that ER stress causes autophagic degradation of Sec63-GFP. Selective autophagy requires the cargo-specific autophagy receptor for recognition and degradation of the cargo [15, 16]. In the budding yeast ER-phagy, Atg39 and Atg40 function as the autophagy receptor [20]. Therefore, we examined whether Atg39 and Atg40 are essential for ER stress-induced autophagic degradation of Sec63-GFP. The *atg39 atg40* double mutations inhibited Sec63-GFP degradation to a similar extent as *atg1* mutation (Fig 1A and S1B Fig). Similar results were obtained using cells deleted for *ATG11*, which encodes an adaptor bridging the autophagosome-formation machinery and autophagy receptors. On the other hand, Sec63-GFP was normally degraded in cells deleting the *ATG32* and *ATG36* genes, which encode an autophagy receptor for mitochondria and peroxisomes, respectively. These results suggest that ER stress triggers autophagic degradation of Sec63-GFP involving Atg39 and Atg40.

We next examined degradation of other ER proteins showing the localization pattern different from Sec63. In the budding yeast, the ER consists of two distinct subdomains, the perinuclear ER and the cortical ER, and these subdomains are bridged by cytoplasmic ER [23]. Sec63 preferentially resides in ER sheets of the perinuclear, cortical, and cytoplasmic ER (S2A Fig). Therefore, we employed the reticulon Rtn1, which enriches in tubules and sheet edges of the ER but is excluded from ER sheets where Sec63 resides (S2A Fig). We also examined the perinuclear ER protein Hmg1 and the inner nuclear membrane protein Src1 (S2A Fig), because the budding yeast perinuclear ER is equivalent to the nuclear envelop. In response to ER stress, free GFP was produced from Rtn1-GFP, Hmg1-GFP and Src1-GFP (S2B–S2D Fig). Production of free GFP from Hmg1-GFP and Src1-GFP were significantly inhibited by *atg1* mutation and *atg39 atg40* double mutations (S2C and S2D Fig), suggesting that autophagy involving Atg39 and Atg40 degrades Hmg1-GFP and Src1-GFP during ER stress response. In contrast, Rtn1-GFP degradation was only partially suppressed by *atg39 atg40* double mutations (S2B Fig). Similar results were obtained using Yop1, which is a reticulon-interacting protein and

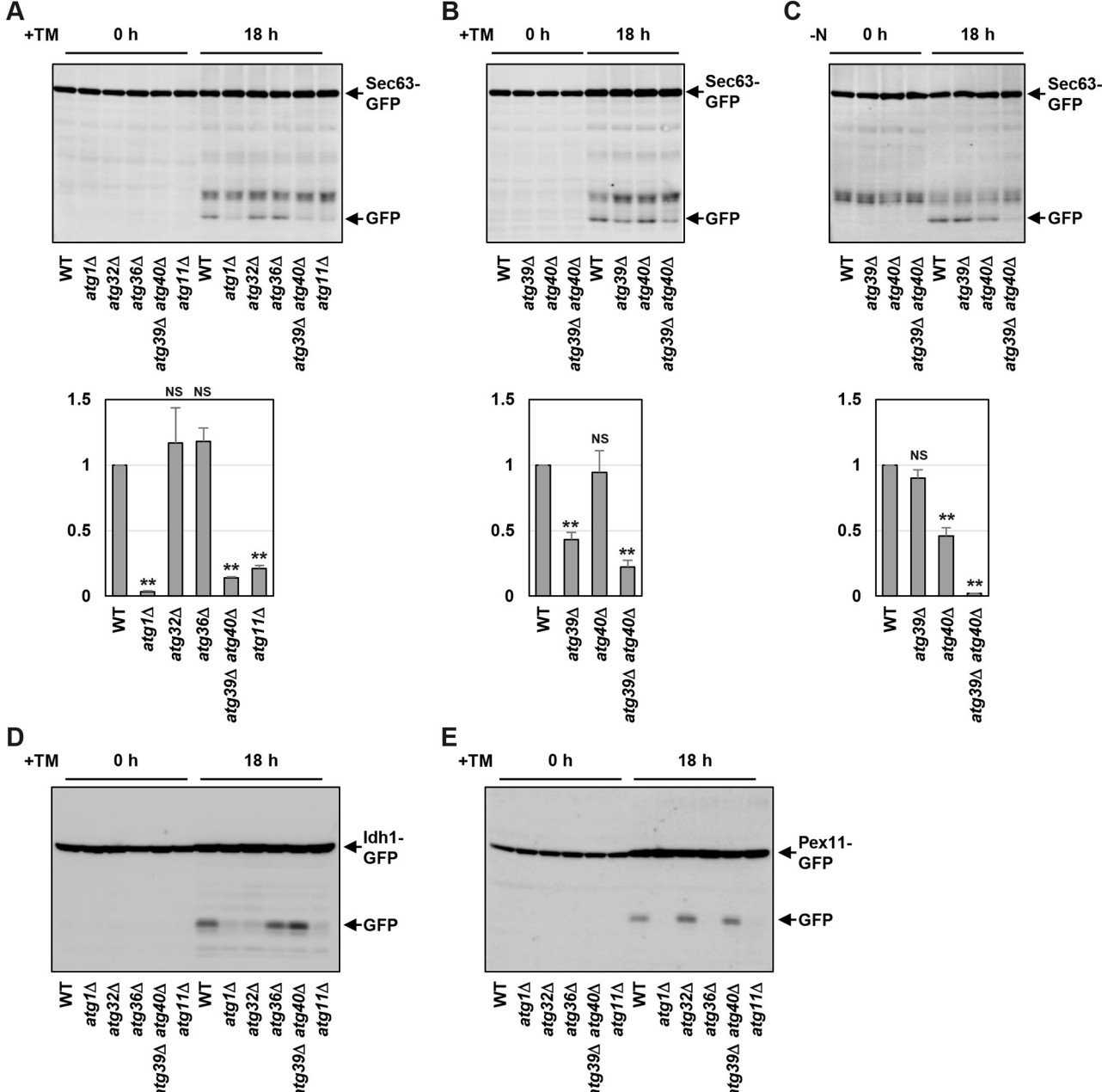

**Fig 1. ER stress activates selective autophagy pathways.** (A, B) Sec63-GFP degradation after ER stress treatment. Wild-type (WT) and indicated mutant strains harboring GFP-tagged *SEC63* were grown at 25 ˚C until exponential phase and treated with 3 μg/ml tunicamycin (TM) for 18 hr. Extracts prepared from each cell were immunoblotted with anti-GFP antibodies. The intensities of free GFP were measured and normalized to the Sec63-GFP level. The values are plotted as the fold change from wild-type cells. The data show mean ± SEM (n = 3). **$P < 0.01$ as determined by Student's *t*-test. NS, not statistically significant ($P > 0.05$). (C) Sec63-GFP degradation after nitrogen starvation. Wild-type (WT) and indicated mutant strains harboring GFP-tagged *SEC63* were grown at 25 ˚C until exponential phase and then incubated under nitrogen-starved conditions for 18 hr. Extracts prepared from each cell were immunoblotted with anti-GFP antibodies. The intensities of free GFP were measured and normalized to the Sec63-GFP level. The values are plotted as the fold change from wild-type cells. The data show mean ± SEM (n = 3). **$P < 0.01$ as determined by Student's *t*-test. NS, not statistically significant ($P > 0.05$). (D) Idh1-GFP degradation after ER stress treatment. Wild-type (WT) and indicated mutant strains harboring GFP-tagged *IDH1* were grown at 25 ˚C until exponential phase and treated with 3 μg/ml tunicamycin (TM) for 18 hr. Extracts prepared from each cell were immunoblotted with anti-GFP antibodies. (E) Pex11-GFP degradation after ER stress treatment. Wild-type (WT) and indicated mutant strains harboring GFP-tagged *PEX11* were grown at 25 ˚C until exponential phase and treated with 3 μg/ml tunicamycin (TM) for 18 hr. Extracts prepared from each cell were immunoblotted with anti-GFP antibodies.

shows the localization pattern similar to Rtn1 (S2A and S2E Fig), suggesting that Rtn1-GFP and Yop1-GFP are degraded by both Atg39/40-dependent and -independent mechanisms. Taken together, these results suggest that ER stress causes ER-phagy involving Atg39 and Atg40.

Next, we examined the relative contributions of Atg39 and Atg40 to ER stress-induced ER-phagy using Sec63-GFP as a marker (Fig 1B). Sec63-GFP degradation in response to ER stress was partly inhibited by *atg39* mutation. The *atg40* mutation alone did not affect ER stress-induced Sec63-GFP degradation but enhanced the *atg39* defect. These results raised the possibility that Atg39 is more important for ER stress-induced ER-phagy than Atg40. To test this, we monitored degradation of Rtn1-GFP, Hmg1-GFP and Src1-GFP. Degradation of these GFP-tagged proteins was more effectively inhibited by *atg39* mutation than *atg40* mutation (S2I–S2K Fig). These results suggest that Atg39 plays a major role in ER stress-induced ER-phagy.

We next investigated whether Atg39 is more important than Atg40 in ER-phagy induced by other stimuli. Sec63-GFP degradation in response to nitrogen starvation was partly and fully blocked in *atg40* single and *atg39 atg40* double mutant cells, respectively, although it was normally occurred in the *atg39* single mutant cells (Fig 1C). Rtn1-GFP degradation induced by nitrogen starvation was prevented by *atg40* mutation, but not by *atg39* mutation (S2L Fig). Thus, Sec63-GFP and Rtn1-GFP are mainly degraded by Atg40 under nitrogen-starved conditions. However, Atg40 seemed to be less important than Atg39 in nitrogen starvation-induced degradation of Hmg1-GFP and Src1-GFP: their degradation upon nitrogen starvation was severely and modestly diminished in *atg39* and *atg40* double mutant cells, respectively (S2M and S2N Fig). Accordingly, the relative functional significance of Atg39 and Atg40 in ER-phagy is changed by the kinds of stimuli.

## Mitophagy and pexophagy are also induced by ER stress

We investigated whether other selective autophagy such as mitophagy and pexophagy are also induced by ER stress. To test this, we used the strains expressing GFP-tagged Idh1 mitochondrial and Pex11 peroxisomal proteins. We could detect free GFP produced from Idh1-GFP and Pex11-GFP in ER-stressed cells (Fig 1D and 1E). The *atg1* mutation significantly reduced free GFP, indicating that ER stress-induced degradation of Idh1-GFP and Pex11-GFP is mediated by autophagy. We further examined the requirement of autophagy receptors for autophagic degradation of Idh1-GFP and Pex11-GFP. Idh1-GFP degradation was blocked in the *atg32* mutant cells, but not in the *atg36* mutant or *atg39 atg40* double mutant cells; Pex11-GFP degradation was blocked in the *atg36* mutant cells, but not in the *atg32* mutant or *atg39 atg40* double mutant cells. These results indicate that ER stress activates not only ER-phagy, but also mitophagy and pexophagy.

## ER stress upregulates the mRNA levels of autophagy receptors

Previous studies provided the evidence that the expression levels of autophagy receptors are upregulated under conditions that induce selective autophagy [17, 19, 20]. Therefore, we measured the mRNA levels of *ATG39*, *ATG40*, *ATG32*, and *ATG36* during ER stress response. The quantitative real-time RT-PCR (qRT-PCR) analysis showed that all these mRNAs increased upon exposure to tunicamycin and DTT (Fig 2). Especially, *ATG39* mRNA levels were dramatically upregulated by ER stress. Thus, ER stress upregulates the mRNA levels of *ATG39*, *ATG40*, *ATG32* and *ATG36*.

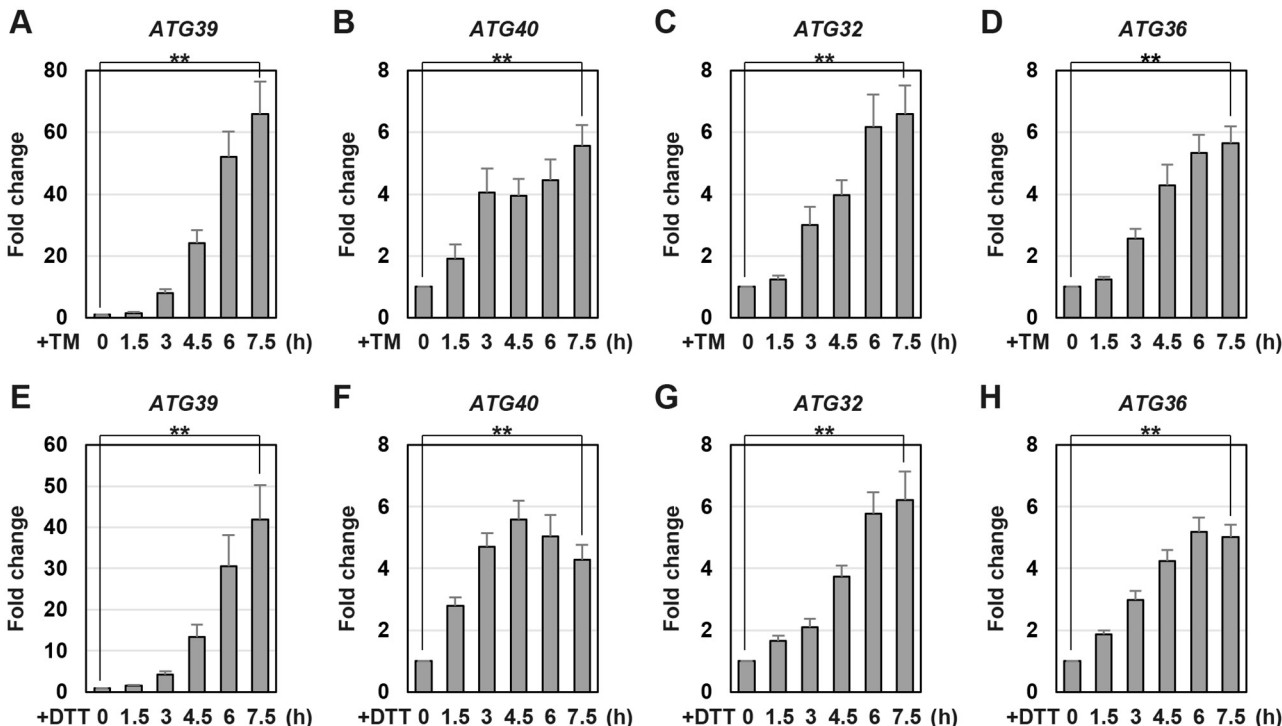

**Fig 2. ER stress upregulates the mRNA levels of the genes encoding a selective autophagy receptor.** The mRNA levels of *ATG39* (A, E), *ATG40* (B, F), *ATG32* (C, G) and *ATG36* (D, H). Wild-type strains were grown at 25 °C until exponential phase and treated with 3 μg/ml tunicamycin (TM) (A-D) or 6 mM dithiothreitol (DTT) (E-H) for the indicated time. The mRNA levels were quantified by qRT-PCR analysis, and relative mRNA levels were calculated using *ACT1* mRNA. The values are plotted as the fold change from untreated cells. The data show mean ± SEM (n > 3). **P < 0.01 as determined by Student's *t*-test.

## Snf1 is involved in *ATG39* induction caused by ER stress

Among mRNAs for autophagy receptors, *ATG39* mRNA was markedly increased (more than 40-fold) by ER stress. This finding led us to use the *ATG39* gene as a model to explore the mechanism by which ER stress induces the expression of autophagy receptors. We attempted to identify the regulator of *ATG39* expression. The UPR pathway composed of Ire1 and Hac1 plays a principal role in transcriptional activation in the budding yeast ER stress response [1, 2]. Therefore, we asked if the UPR pathway activates the *ATG39* gene. However, *ATG39* mRNA was normally increased in the *hac1* mutant cells (S3A Fig). In the budding yeast, stress responsive MAPKs, such as Mpk1 and Hog1, and Snf1 AMPK are activated by ER stress [4–12]. We therefore tested whether they are involved in *ATG39* upregulation. In *mpk1* and *hog1* mutant cells, *ATG39* induction was unaffected (S3B and S3C Fig). In contrast, *ATG39* induction was decreased in the *snf1* mutant cells (Fig 3A and S4A Fig). Consistent with the mRNA levels, Atg39 protein levels were upregulated by ER stress, and this upregulation was suppressed by *snf1* mutation (Fig 3B). It should be noted that a slower migrating form of Atg39 in SDS-polyacrylamide gel electrophoresis (SDS-PAGE) appeared after exposure to ER stress and this form was still observed in the extract treated with alkali phosphatase (S5A and S5B Fig), implying that ER stress causes Atg39 posttranslational modification except for phosphorylation. To further investigate the involvement of Snf1 in *ATG39* expression, we employed the *reg1* mutant cells where Snf1 is hyperactivated [10, 24, 25]. The *reg1* mutation elevated *ATG39* mRNA levels under both unstressed and ER-stressed conditions (Fig 3C). Furthermore, this *reg1* phenotype was suppressed by *snf1* mutation. These results strongly support that Snf1

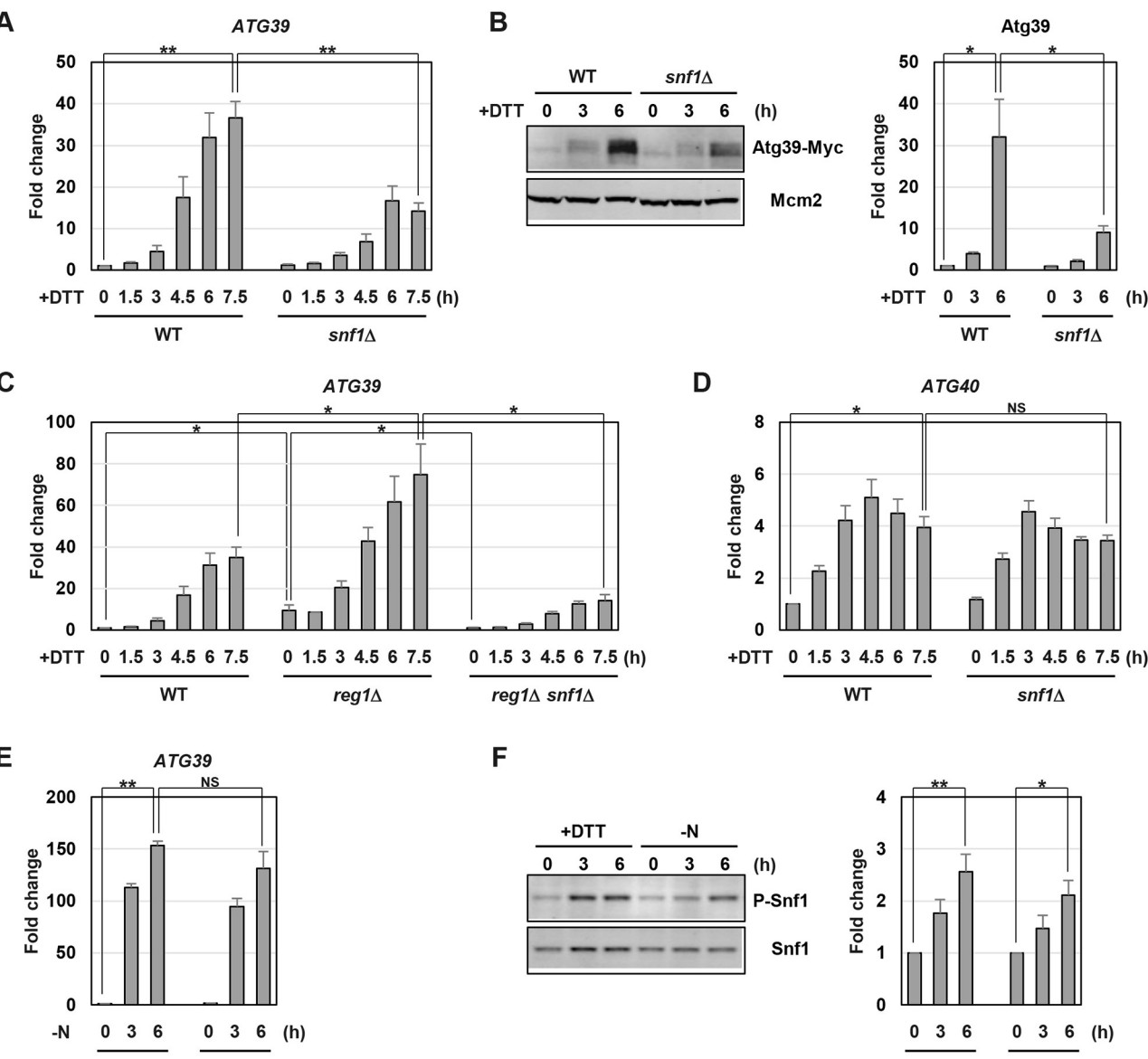

**Fig 3. Snf1 is involved in ER stress-induced *ATG39* upregulation.** (A) The *ATG39* mRNA levels in ER-stressed *snf1* mutant. Wild-type (WT) and *snf1* mutant strains were grown at 25 ˚C until exponential phase and treated with 6 mM dithiothreitol (DTT) for the indicated time. The *ATG39* mRNA levels were quantified by qRT-PCR analysis, and relative mRNA levels were calculated using *ACT1* mRNA. The values are plotted as the fold change from wild-type cells at the time of DTT addition. The data show mean ± SEM (n = 3). **$P < 0.01$ as determined by Student's *t*-test. (B) The Atg39 protein level in ER-stressed *snf1* mutant. Wild-type (WT) and *snf1* mutant strains harboring Myc-tagged *ATG39* were grown at 25 ˚C until exponential phase and treated with 6 mM dithiothreitol (DTT) for the indicated time. Extracts prepared from each cell were immunoblotted with anti-Myc antibodies. The intensities of Atg39-Myc were measured and normalized to the Mcm2 level. The values are plotted as the fold change from wild-type cells at the time of DTT addition. The data show mean ± SEM (n = 4). *$P < 0.05$ as determined by Student's *t*-test. (C) The *ATG39* mRNA levels in the Snf1-activated cells. Wild-type (WT) and indicated mutant strains were grown at 25 ˚C until exponential phase and treated with 6 mM dithiothreitol (DTT) for the indicated time. The *ATG39* mRNA levels were quantified by qRT-PCR analysis, and relative mRNA levels were calculated using *ACT1* mRNA. The values are plotted as the fold change from wild-type cells at the time of DTT addition. The data show mean ± SEM (n = 3). *$P < 0.05$ as determined by Student's *t*-test. (D) The *ATG40* mRNA levels in ER-stressed *snf1* mutant. Wild-type (WT) and *snf1* mutant strains were grown at 25 ˚C until exponential phase and treated with 6 mM dithiothreitol (DTT) for the indicated time. The *ATG40* mRNA levels were quantified by qRT-PCR analysis, and relative mRNA levels were calculated using *ACT1* mRNA. The values are plotted as the fold change from wild-type cells at the time of DTT addition. The data show mean ± SEM (n = 3). *$P < 0.05$ as determined by Student's *t*-test. NS, not statistically significant ($P > 0.05$). (E) The *ATG39* mRNA levels in nitrogen-starved *snf1* mutant. Wild-type (WT) and *snf1* mutant strains were grown at 25 ˚C until exponential phase and then incubated under nitrogen starvation conditions for the indicated time. The *ATG39* mRNA levels were quantified by qRT-PCR analysis, and relative mRNA levels were calculated using *ACT1* mRNA. The values are plotted as the fold change from wild-type cells at the time of nitrogen removal. The data show mean ± SEM (n = 3). **$P < 0.01$ as determined by Student's *t*-test. NS, not statistically significant ($P > 0.05$). (F) Effects of nitrogen

starvation on Snf1 activation. The wild-type cells were grown at 25 ˚C until exponential phase and treated with 6 mM dithiothreitol (DTT) or incubated under nitrogen-starved conditions for the indicated time. Extracts prepared from each cell were immunoblotted with anti-phospho-AMPKα (P-Snf1) and anti-Snf1 antibodies. The intensities of phosphorylated Snf1 were measured and normalized to total Snf1 level. The values are plotted as the fold change from untreated cells. The data show mean ± SEM (n = 4). $^*P < 0.05$ and $^{**}P < 0.01$ as determined by Student's $t$-test.

positively regulates *ATG39* expression in ER stress response. To ask whether Snf1 specifically regulates *ATG39* expression, we monitored the mRNA levels of other autophagy receptors following exposure to ER stress. *ATG40*, *ATG32* and *ATG36* were normally induced in the *snf1* mutant cells (Fig 3D and S4B–SD Fig), suggesting that their induction is mediated independently of Snf1.

We examined whether Snf1 regulates *ATG39* expression under other conditions which induce ER-phagy. The protein levels of Atg39 are elevated when cells are treated with rapamycin, a compound that mimics nitrogen starvation [20]. Therefore, we tested whether *ATG39* mRNA levels are increased by cultivation in the nitrogen starvation medium. We found that nitrogen starvation caused a dramatic increase in *ATG39* mRNA (Fig 3E). However, *ATG39* induction by nitrogen starvation was largely unaffected by *snf1* mutation. These results suggest that during nitrogen starvation, *ATG39* expression is upregulated in a Snf1-independent manner. The observation that *ATG39* upregulation by nitrogen starvation normally occurred in the *snf1* mutant cells led us to hypothesize that unlike ER-stressed conditions, Snf1 activity remains unchanged under nitrogen-starved conditions. Therefore, we examined the effect of nitrogen starvation on Snf1 activity. Snf1 is a budding yeast ortholog of mammalian AMPKα [24, 25]. Antibodies against the phosphorylated (activated) form of mammalian AMPKα can be utilized to detect activated Snf1 [10, 26]. We performed western blot analysis using anti-phospho-AMPKα antibodies and found that Snf1 activity was upregulated under nitrogen starved-conditions (Fig 3F). These results suggest that the mechanism independent of Snf1 mainly operates in *ATG39* induction by nitrogen starvation.

## Mig1 and Mig2 negatively regulate *ATG39* expression

We explored the mechanism by which Snf1 induces *ATG39* expression. Previous studies revealed that a variety of transcription factors are phosphorylated and regulated by Snf1 [24, 25], raising the possibility that Snf1 leads to activation of the *ATG39* promoter. To test this, we constructed the $P_{ATG39}$-*GFP* reporter, consisting of the 5' upstream region of the *ATG39* gene to drive *GFP* expression. *GFP* expression from the $P_{ATG39}$-*GFP* reporter was increased after exposure to ER stress, and this induction was significantly diminished by *snf1* mutation (Fig 4A). These results suggest that Snf1 is involved in transcriptional activation of the *ATG39* gene during ER stress response.

We next attempted to identify the factor that acts downstream of Snf1 and regulates *ATG39* expression. Snf1 phosphorylates and regulates many transcription factors, including Mig1, Mig2, Hsf1, Cat8, Sip1, Adr1, Msn2, Rds2, Gcn4, Rgt1 and Gln3 [24, 25]. To investigate whether these transcription factors are involved in *ATG39* expression, we utilized *reg1* mutation which increases *ATG39* mRNA levels under unstressed conditions in a Snf1-dependent manner (Fig 3C). If a given transcription factor positively regulates *ATG39* expression, its mutation should suppress *ATG39* upregulation caused by *reg1* mutation, similar to *snf1* mutation. Conversely, if a given transcription factor negatively regulates *ATG39* expression, its mutation should lead to *ATG39* upregulation, similar to *reg1* mutation. Therefore, we examined whether mutations in candidate genes impact *ATG39* mRNA levels under unstressed conditions in the absence or presence of the *REG1* gene. However, none of their single mutation markedly increased *ATG39* expression in a *REG1* background nor suppressed *ATG39*

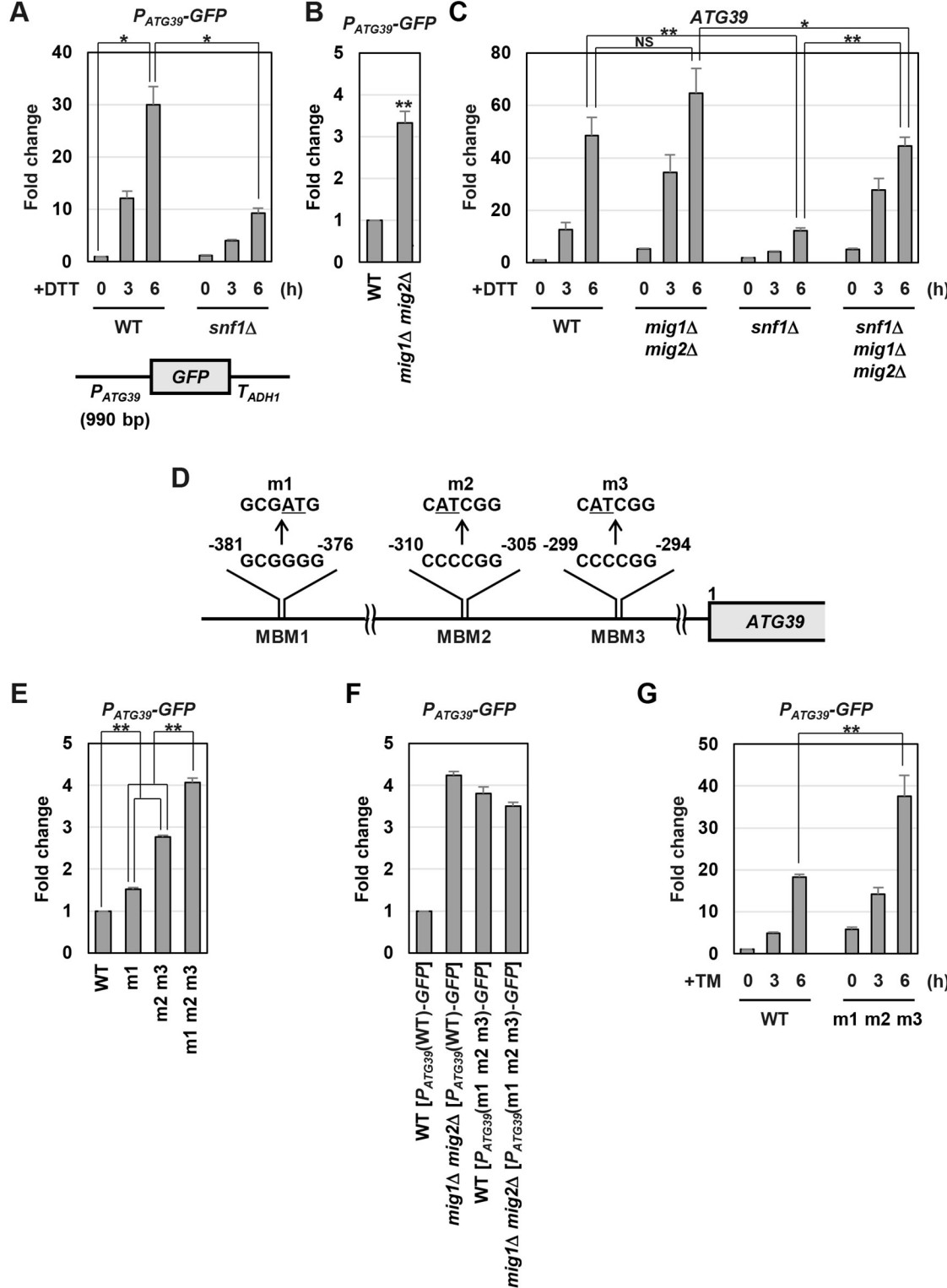

**Fig 4. Mig1/2 repress *ATG39* expression.** (A) Effects of ER stress and *snf1* mutation on expression of $P_{ATG39}$-*GFP* reporter. Wild-type (WT) and *snf1* mutant strains harboring the integration which expresses GFP under the control of the *ATG39* promoter were grown at 25 °C until exponential phase and treated with 6 mM dithiothreitol (DTT) for the indicated time. The *GFP* mRNA levels were quantified by qRT-PCR analysis, and relative mRNA levels were calculated using *ACT1* mRNA. The values are plotted as the fold change from wild-type cells at the time of DTT addition. The data show mean ± SEM (n = 3). $^*P < 0.05$ as determined by

Student's *t*-test. Schematic representation of the structure of $P_{ATG39}$-*GFP* is also shown. (B) Effects of *mig1 mig2* mutations on expression of $P_{ATG39}$-*GFP* reporter. Wild-type (WT) and *mig1 mig2* mutant strains harboring the integration which expresses GFP under the control of the *ATG39* promoter were grown at 25 ˚C until exponential phase. The *GFP* mRNA levels were quantified by qRT-PCR analysis, and relative mRNA levels were calculated using *ACT1* mRNA. The values are plotted as the fold change from wild-type cells. The data show mean ± SEM (n = 3). $^{**}P < 0.01$ as determined by Student's *t*-test. (C) The *ATG39* mRNA levels in ER-stressed *mig1 mig2* mutant. Wild-type (WT) and indicated mutant strains were grown at 25 ˚C until exponential phase and treated with 6 mM dithiothreitol (DTT) for the indicated time. The *ATG39* mRNA levels were quantified by qRT-PCR analysis, and relative mRNA levels were calculated using *ACT1* mRNA. The values are plotted as the fold change from wild-type cells at the time of DTT addition. The data show mean ± SEM (n = 3). $^{*}P < 0.05$ and $^{**}P < 0.01$ as determined by Student's *t*-test. NS, not statistically significant ($P > 0.05$). (D) Three putative Mig1/2-binding motifs in *ATG39* promoter region. (E) Effects of mutations in putative Mig1/2-binding motifs on expression of $P_{ATG39}$-*GFP* reporter. Wild-type (WT) cells harboring the integration which expresses GFP under the control of wild-type or mutated *ATG39* promoter were grown at 25 ˚C until exponential phase. The *GFP* mRNA levels were quantified by qRT-PCR analysis, and relative mRNA levels were calculated using *ACT1* mRNA. The values are plotted as the fold change from wild-type cells. The data show mean ± SEM (n = 3). $^{**}P < 0.01$ as determined by Student's *t*-test. (F) Effects of mutations in putative Mig1/2-binding motifs and *MIG1/2* genes on expression of $P_{ATG39}$-*GFP* reporter. Wild-type (WT) and *mig1 mig2* mutant strains harboring the integration which expresses GFP under the control of wild-type or mutated *ATG39* promoter were grown at 25 ˚C until exponential phase. The *GFP* mRNA levels were quantified by qRT-PCR analysis, and relative mRNA levels were calculated using *ACT1* mRNA. The values are plotted as the fold change from wild-type cells harboring the integration which expresses GFP under the control of wild-type *ATG39* promoter. The data show mean ± SEM (n = 4). (G) Effects of mutations in putative Mig1/2-binding motifs on ER stress-induced *ATG39* upregulation. Wild-type (WT) cells harboring the integration which expresses GFP under the control of wild-type or mutated *ATG39* promoter were grown at 25 ˚C until exponential phase and treated with 3 μg/ml tunicamycin (TM) for the indicated time. The *GFP* mRNA levels were quantified by qRT-PCR analysis, and relative mRNA levels were calculated using *ACT1* mRNA. The values are plotted as the fold change from wild-type cells harboring the integration which expresses GFP under the control of wild-type *ATG39* promoter at the time of TM addition. The data show mean ± SEM (n = 4). $^{**}P < 0.01$ as determined by Student's *t*-test.

upregulation caused by *reg1* mutation (S6 Fig). Among transcription factors known to be regulated by Snf1, Mig1 and Mig2 are transcriptional repressors closely related to each other. Indeed, they redundantly repress some glucose-repressed genes [27, 28]. Therefore, we examined whether Mig1 and Mig2 act redundantly to repress *ATG39* expression. The *mig1 mig2* double mutations significantly increased *ATG39* expression under unstressed conditions, but neither *mig1* nor *mig2* single mutation did (S6C Fig). *GFP* expression from the $P_{ATG39}$-*GFP* reporter was also increased by *mig1 mig2* double mutations (Fig 4B). These results suggest that Mig1 and Mig2 redundantly repress *ATG39* promoter activity under unstressed conditions. We next examined the involvement of Mig1 and Mig2 in *ATG39* induction caused by ER stress. The *mig1 mig2* double mutations upregulated *ATG39* induction upon exposure to tunicamycin (S7A Fig), although the difference in *ATG39* expression levels between wild-type and *mig1 mig2* double mutant cells did not reach statistical significance when DTT was used as an ER stressor (Fig 4C). Furthermore, reduced *ATG39* induction observed in the *snf1* mutant cells was clearly suppressed by *mig1 mig2* double mutations. These results suggest that *ATG39* expression is inhibited by Mig1 and Mig2, and this inhibition is itself negatively regulated by Snf1. Furthermore, it is suggested that an additional molecule may function downstream of Snf1 to regulate *ATG39* expression, since *snf1* mutation downregulated *ATG39* induction caused by ER stress even in the *mig1 mig2* double mutant cells.

We further explored the mechanism by which Mig1 and Mig2 control *ATG39* promoter activity. Previous analysis revealed that Mig1 binds the consensus sequence, GCGGGG [29]. Our sequence analysis utilizing JASPAR, a database of transcription factor binding profiles (http://jaspar.genereg.net/), showed that three putative binding motifs for Mig1 and Mig2 exist in the 5' upstream region of the *ATG39* gene (Fig 4D). We designated them as the Mig1/2-binding motifs (MBMs). To examine whether the MBMs are important for transcriptional regulation of *ATG39*, we mutated MBMs in the $P_{ATG39}$-*GFP* reporter. Mutation of MBM1, the sequence spanning from -381 to -376, slightly increased *GFP* expression; double mutations of MBM2 and MBM3, the sequences spanning from -310 to -305 and from -299 to -294, respectively, modestly increased *GFP* expression; triple mutations of MBMs further elevated the

expression level of *GFP* (Fig 4E). *GFP* expression from the P*ATG39*-*GFP* reporter containing triple MBM mutations was not enhanced by *mig1 mig2* double mutations (Fig 4F). Furthermore, triple MBM mutations potentiated *GFP* induction after exposure to tunicamycin (Fig 4G), although the difference in *GFP* expression levels between wild-type and mutated promoters did not reach statistical significance when DTT was used as an ER stressor (S7B Fig). Thus, the MBMs function in cis to mediate repression of the *ATG39* gene. These results strongly support the model that Mig1 and Mig2 negatively regulate *ATG39* through the MBMs in its promoter.

## Snf1 promotes nuclear export of Mig1/2 in ER stress response

Our previous findings that Snf1 activity is enhanced by ER stress [10] suggested that Snf1-mediated phosphorylation of Mig1 and Mig2 may be increased by ER stress. Phosphorylation of Mig1 and Mig2 could be detected by alterations of mobility in SDS-PAGE [30, 31]. Therefore, we analyzed mobility of the carboxyl-terminally Myc-tagged Mig1 in SDS-PAGE (Fig 5A). Slower migrating forms of Mig1 were observed in wild-type cells under unstressed conditions and increased after ER stress treatment. Slower migration of Mig1 seen in ER-stressed cells could not be detected by treatment of extracts with alkali phosphatase, indicating that the reduced mobility of Mig1 is caused by phosphorylation (S8C Fig). Furthermore, *snf1* mutation decreased slower migrating Mig1 under unstressed conditions and abolished its increase in response to ER stress. Similar results were obtained using the carboxyl-terminally Myc-tagged Mig2 (Fig 5B). These results suggest that Mig1 and Mig2 are phosphorylated by Snf1 under unstressed conditions and Snf1-dependent phosphorylation of Mig1 and Mig2 is upregulated by ER stress.

Phosphorylation by Snf1 promotes nuclear export of Mig1 and Mig2 [24, 25, 31]. Therefore, we examined whether the cellular localization of Mig1 and Mig2 is altered by ER stress using the strains that express the carboxyl-terminally GFP-tagged Mig1 and Mig2. Mig1-GFP accumulated in the nucleus under unstressed conditions, and this accumulation was suppressed by ER stress (Fig 5C). The *snf1* mutation increased nuclear-localized Mig1 under both unstressed and ER-stressed conditions. Mig2 also accumulated in the nucleus under unstressed conditions (Fig 5D). ER stress significantly reduced nuclear-localized Mig2, which was clearly inhibited in the *snf1* mutant cells. These results suggest that Snf1 promotes nuclear export of Mig1 and Mig2 in ER stress response.

To further investigate the effects of Snf1-mediated phosphorylation, we constructed a phospho-defective form of Mig1, hereafter called Mig1(SATA) [30]. In Mig1(SATA), 7 residues of serine or threonine (S222, S278, S310, S311, S379, T380 and S381), including potential Snf1 recognition sites and their adjacent phosphorylatable residues, are mutated into alanine. We initially assessed the impact of these amino acid substitutions on Mig1 localization. Wild-type Mig1 predominantly localized to the nucleus in a *REG1* background but distributed throughout the cytoplasm and nucleus in a *reg1* mutant background (S8D Fig). This *reg1* phenotype was suppressed by *snf1* mutation, indicating that Snf1 hyperactivation caused by *reg1* mutation facilitates phosphorylation and its consequent nuclear export of wild-type Mig1. Accordingly, a phospho-defective Mig1 was expected to localize to the nucleus even in a *reg1* mutant background. Indeed, nuclear accumulation of Mig1(SATA) in a *reg1* mutant background was observed (S8E Fig), indicating that Mig1(SATA) acts as a phospho-defective mutant. We next examined the cellular localization of Mig1(SATA) during ER stress response. Compared with wild-type Mig1, Mig1(SATA) strongly localized in the nucleus under both unstressed and ER-stressed conditions (Fig 5E). This observation supports that the model in which Snf1 is activated by ER stress and mediates Mig1 phosphorylation and nuclear export. However, nuclear-localized Mig1 was decreased by ER stress even in a *snf1* mutant background and when Mig1

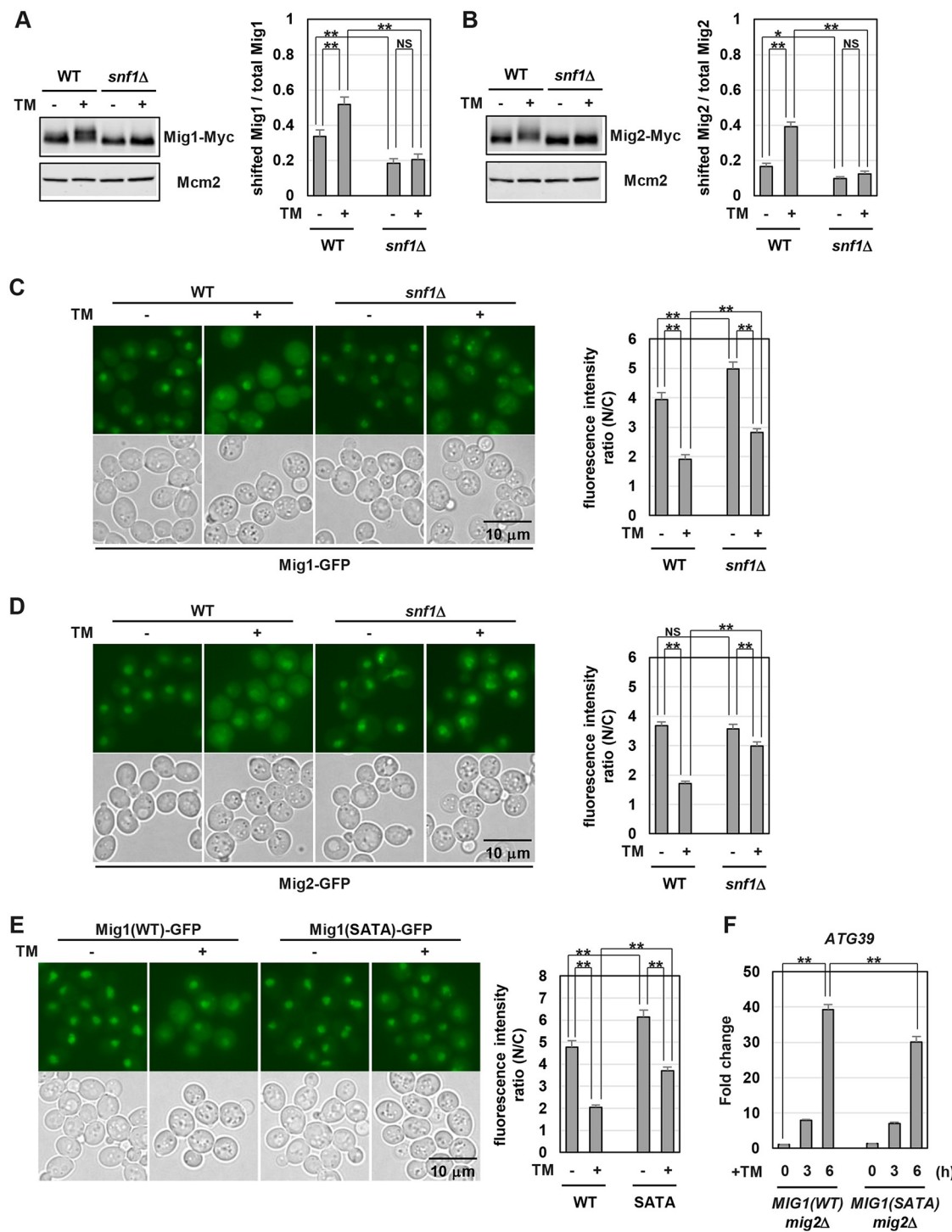

**Fig 5. Nuclear localization of Mig1/2 is suppressed by ER stress.** (A, B) The electrophoretic mobility patterns of Mig1 (A) and Mig2 (B). Wild-type (WT) and *snf1* mutant strains harboring Myc-tagged *MIG1* (A) or *MIG2* (B) were grown at 25 ˚C until exponential phase and treated with 3 μg/ml tunicamycin (TM) for 3 hr. Extracts prepared from each cell were immunoblotted with anti-Myc antibodies. The intensities of shifted Mig1-Myc and Mig2-Myc were measured, and then the ratios of the intensity of shifted Mig1-Myc and Mig2-Myc/that of total Mig1-Myc and Mig2-Myc were calculated, respectively. The data show mean ± SEM (n = 4). *P < 0.05 and **P < 0.01 as determined by Student's *t*-test. NS, not statistically significant (P > 0.05). The protein levels of Mig1 and Mig2 are shown in S8A and S8B Fig, respectively. (C, D) Cellular localization of Mig1 (C) and Mig2 (D). Wild-type (WT) and *snf1* mutant strains harboring GFP-tagged *MIG1* (C) or *MIG2* (D) were grown at 25 ˚C until exponential phase, treated with 3 μg/ml tunicamycin (TM) for 3 hr, and subjected to microscopy. The fluorescence intensities were measured, and then the ratios

(N/C) of the fluorescence intensity per unit area in the nucleus/that in the cytoplasm were calculated. The graphs show mean ± SEM (n = 30). **$P < 0.01$ as determined by Student's *t*-test. NS, not statistically significant ($P > 0.05$). Scale bar, 10 µm. (E) Cellular localization of the phospho-defective form of Mig1. Wild-type (WT) strains harboring GFP-tagged *MIG1* were grown at 25 °C until exponential phase, treated with 3 µg/ml tunicamycin (TM) for 3 hr, and subjected to microscopy. The fluorescence intensities were measured, and then the ratios (N/C) of the fluorescence intensity per unit area in the nucleus/that in the cytoplasm were calculated. The graphs show mean ± SEM (n = 30). **$P < 0.01$ as determined by Student's *t*-test. Scale bar, 10 µm. (F) Effects of the phospho-defective mutation of Mig1 on ER stress-induced *ATG39* upregulation. The *mig2* mutant strains harboring Myc-tagged wild-type or the phospho-defective mutant *MIG1* were grown at 25 °C until exponential phase and treated with 3 µg/ml tunicamycin (TM) for the indicated time. The *ATG39* mRNA levels were quantified by qRT-PCR analysis, and relative mRNA levels were calculated using *ACT1* mRNA. The values are plotted as the fold change from *MIG1(WT) mig2* cells at the time of TM addition. The data show mean ± SEM (n = 4). **$P < 0.01$ as determined by Student's *t*-test.

(SATA) was used (Fig 5C and 5E). These results suggest that a Snf1-independent mechanism may regulate the cellular localization of Mig1 during ER stress response. We also attempted to generate a phospho-mimetic of Mig1 by substitution of potential Snf1 phosphorylation sites with acidic amino acids. However, Mig1(SETE), in which S222, S278, S310, S311, S379, T380 and S381 are mutated into glutamate, displayed the localization pattern similar to Mig1 (SATA) (S8E Fig). Thus, Mig1(SETE) does not act as a phospho-mimetic, but rather behaves as a phospho-defective mutant.

The findings that Mig1(SATA) strongly localized in the nucleus compared with wild-type Mig1 raised the possibility that Mig1(SATA) functions as a stronger repressor than wild-type Mig1. To test this, we utilized *reg1* mutation which increases *ATG39* mRNA under unstressed conditions. We also deleted the *MIG2* gene to exclude contribution of Mig2 to *ATG39* expression. *ATG39* upregulation caused by *reg1* mutation was partly but significantly suppressed by Mig1(SATA) (S8F Fig). Furthermore, we found that *ATG39* induction after ER stress treatment was downregulated by Mig1(SATA) (Fig 5F). These results not only suggest that Snf1-mediated Mig1 phosphorylation is important for the control of *ATG39* expression, but also imply the existence of additional molecules that regulate *ATG39* expression downstream of Snf1.

## Snf1 positively and Mig1/2 negatively regulate ER stress-induced ER-phagy

We asked whether Snf1 regulates ER stress-induced ER-phagy. Sec63-GFP degradation in response to ER stress was modestly diminished in the *snf1* mutant cells (Fig 6A). Similarly, *snf1* mutation reduced degradation of Hmg1-GFP (S9G Fig). In contrast, *reg1* mutation facilitated Sec63-GFP degradation in response to ER stress (S9A Fig). This *reg1* phenotype was suppressed by *snf1* mutation. These results suggest that Snf1 positively regulates ER stress-induced ER-phagy. To investigate the possibility that *snf1* mutation generally reduces autophagic activities during ER stress response, we monitored non-selective autophagy using strains that express the cytoplasmic Pgk1 fused to GFP [32]. Autophagic degradation of Pgk1-GFP in the *snf1* mutant cells was comparable to that in wild-type cells (S9B Fig). Thus, Snf1 appears to be dispensable for activation of non-selective autophagy in response to ER stress. We also examined whether Snf1 participates in nitrogen starvation-induced ER-phagy. Nitrogen starvation caused Sec63-GFP degradation in the *snf1* mutant cells at a level comparable to that in wild-type cells (Fig 6B). Thus, Snf1 is specifically involved in activation of ER-phagy in response to ER stress. We next examined the effect of *snf1* mutation in cells lacking Atg39 or Atg40. In an *atg39* mutant background, *snf1* mutation had no effect on Sec63-GFP degradation (S9C Fig); in contrast, in an *atg40* mutant background, *snf1* mutation significantly reduced Sec63-GFP degradation (S9D Fig). These results suggest that Snf1 regulates ER stress-induced ER-phagy via Atg39.

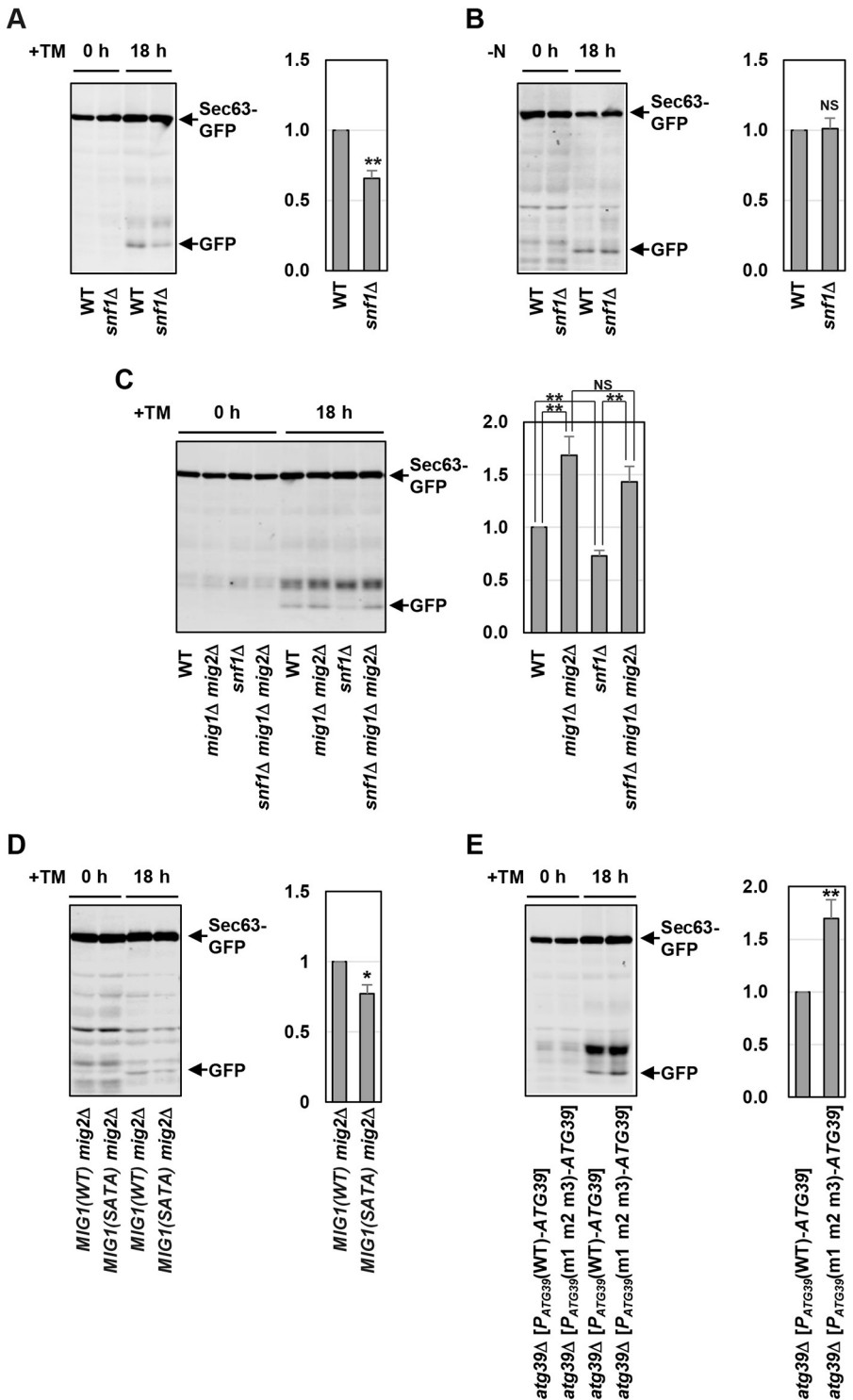

**Fig 6. Snf1 and Mig1/2 regulate ER stress-induced ER-phagy.** (A) Sec63-GFP degradation in ER-stressed *snf1* mutant. Wild-type (WT) and *snf1* mutant strains harboring GFP-tagged *SEC63* were grown at 25 ˚C until exponential phase and treated with 3 μg/ml tunicamycin (TM) for 18 hr. Extracts prepared from each cell were immunoblotted with anti-GFP antibodies. The intensities of free GFP were measured and normalized to the Sec63-GFP level. The values are plotted as the fold change from wild-type cells. The data show mean ± SEM (n = 5). \*\**P* < 0.01 as determined by Student's *t*-test. (B) Sec63-GFP degradation in nitrogen-starved *snf1* mutant. Wild-type (WT) and *snf1* mutant strains harboring GFP-tagged *SEC63* were grown at 25 ˚C until exponential phase and then incubated under

nitrogen starvation conditions for 18 hr. Extracts prepared from each cell were immunoblotted with anti-GFP antibodies. The intensities of free GFP were measured and normalized to the Sec63-GFP level. The values are plotted as the fold change from wild-type cells. The data show mean ± SEM (n = 4). NS, not statistically significant (*P* > 0.05), as determined by Student's *t*-test. (C) Sec63-GFP degradation in *mig1 mig2* mutant. Wild-type (WT) and indicated mutant strains harboring GFP-tagged *SEC63* were grown at 25 ˚C until exponential phase and treated with 3 μg/ml tunicamycin (TM) for 18 hr. Extracts prepared from each cell were immunoblotted with anti-GFP antibodies. The intensities of free GFP were measured and normalized to the Sec63-GFP level. The values are plotted as the fold change from wild-type cells. The data show mean ± SEM (n = 3). **P* < 0.01 as determined by Student's *t*-test. NS, not statistically significant (*P* > 0.05). (D) Effects of the phospho-defective mutation of Mig1 on Sec63-GFP degradation. The *mig2* mutant strains harboring GFP-tagged *SEC63* and Myc-tagged wild-type or the phospho-defective mutant *MIG1* were grown at 25 ˚C until exponential phase and treated with 3 μg/ml tunicamycin (TM) for 18 hr. Extracts prepared from each cell were immunoblotted with anti-GFP antibodies. The intensities of free GFP were measured and normalized to the Sec63-GFP level. The values are plotted as the fold change from *MIG1(WT) mig2* cells. The data show mean ± SEM (n = 4). **P* < 0.05 as determined by Student's *t*-test. (E) Sec63-GFP degradation in cells expressing *ATG39* under the control of mutated *ATG39* promoter. The *atg39* mutants harboring GFP-tagged *SEC63* and the integration which expresses *ATG39* under the control of wild-type or mutated *ATG39* promoter were grown at 25 ˚C until exponential phase and treated with 3 μg/ml tunicamycin (TM) for 18 hr. Extracts prepared from each cell were immunoblotted with anti-GFP antibodies. The intensities of free GFP were measured and normalized to the Sec63-GFP level. The values are plotted as the fold change from cells harboring the wild-type *ATG39* integration. The data show mean ± SEM (n = 4). **P* < 0.01 as determined by Student's *t*-test.

Next, we investigated whether Mig1 and Mig2 regulate ER stress-induced ER-phagy. Sec63-GFP degradation in response to ER stress was significantly increased in the *mig1 mig2* double mutant cells (Fig 6C). Furthermore, reduced Sec63-GFP degradation in the *snf1* mutant cells was suppressed by *mig1 mig2* double mutations. Similar results were obtained using Hmg1-GFP as a marker (S9G Fig). However, degradation of Sec63-GFP and Hmg1-GFP could not be detected in the *mig1 mig2* double mutant cells under unstressed conditions (Fig 6C, and S9E and S9G Fig). This is consistent with the observation that GFP-Atg8 degradation, a typical indicator for activation of the core autophagy machinery, was also undetectable in unstressed *mig1 mig2* double mutant cells (S9F Fig) and suggests that upregulation of *ATG39* expression alone is not enough to induce ER-phagy. To test the importance of Snf1-mediated Mig1 phosphorylation in ER stress-induced ER-phagy, we used cells which express a phospho-defective form of Mig1, Mig1(SATA). In the cells expressing Mig1(SATA), ER stress-induced degradation of Sec63-GFP and Hmg1-GFP was decreased compared with the cells expressing wild-type Mig1 (Fig 6D and S9H Fig). Taken together, these results suggest that Mig1 and Mig2 function downstream of Snf1 to negatively regulate ER stress-induced ER-phagy.

We next tested whether transcriptional repression of the *ATG39* gene by Mig1 and Mig2 is important for regulation of ER stress-induced ER-phagy. To test this, we generated strains expressing Atg39 under the control of wild-type or mutated *ATG39* promoter. Degradation of Sec63-GFP and Hmg1-GFP in response to ER stress was significantly increased when Atg39 was expressed from the *ATG39* promoter mutated in MBMs (Fig 6E and S9I Fig). These results suggest that the control of *ATG39* promoter activity via MBMs is critical for regulation of ER stress-induced ER-phagy.

Finally, we investigated the physiological role of ER-phagy in ER stress response. We tested ER stress sensitivity of cells deleted for *ATG39*, *ATG40* or both. Cells were plated on medium containing tunicamycin as an inducer of ER stress, and their growth was monitored. However, none of them exhibited an obvious ER stress sensitive phenotype (S10A Fig). We also examined ER stress sensitivity in a *snf1* mutant background. As shown in our previous paper [10], the *snf1* mutant cells were resistant to ER stress (S10B Fig). This *snf1* phenotype was not modified by *atg39* and *atg40* mutations. Thus, activation of ER-phagy is not essential for cellular tolerance to ER stress.

## Discussion

### ER-phagy is induced by ER stress

In this study, we observed ER degradation mediated by macroautophagy during ER stress response, although a previous report could not detect it [14]. This difference is probably due to different experimental conditions. In the previous report, Sec63-GFP degradation was examined within a short period after ER stress treatment. Consistent with the previous report, autophagic degradation of Sec63-GFP was hardly detected within 6 hours in our experiments. However, we detected Sec63-GFP degradation 18 hours after exposure to ER stressors. Selective macroautophagy is normally analyzed by detection of free GFP that is produced by degradation of GFP-tagged protein specifically localized. This analysis utilizes the properties of GFP that is resistant to the vacuole-resident proteases and requires the time course to accumulate free GFP. Our results show that the long-term cultivation in the presence of ER stressors is needed to detect ER degradation caused by ER stress.

A previous study showed that the *atg39* mutant cells are sensitive to nitrogen starvation [20]. In contrast, we found that a deficiency in ER-phagy caused by *atg39 atg40* double mutations has no effect on the susceptibility to ER stress. Deletion of the core autophagy-related genes does not result in hypersensitivity to ER stress [14]. Hence, the physiological significance of ER-phagy in ER stress response remains unclear. One plausible explanation for why a defect in ER stress-induced ER-phagy fails to cause ER stress hypersensitivity is that ER degradation occurs in a manner independent of ER-phagy. Indeed, microautophagy of the ER is induced by ER stress [14]. Additionally, ERAD is potentiated through transcriptional activation of genes involved in it during ER stress response [33]. Thus, multiple pathways function in ER degradation during ER stress response. Alternatively, as assumed in mammalian SEC62-mediating ER-phagy [34], yeast ER-phagy induced by ER stress may function to decrease the ER size once expanded by ER stress, but not be needed for elimination of aberrant proteins in the ER.

We unexpectedly found that ER stress triggers not only ER-phagy but also mitophagy and pexophagy. Induction of selective autophagy requires activation of the core autophagy machinery and expression of the cargo-specific autophagy receptor [15, 16]. It has been demonstrated that the core autophagy machinery becomes activated in the budding yeast ER stress response [13]. Furthermore, we found here that ER stress increases expression of the *ATG32* and *ATG36* genes encoding an autophagy receptor for mitophagy and pexophagy, respectively. Not only that ER stress inhibits TORC1 but also that TORC1 negatively regulates *ATG32* expression have been shown previously [35, 36]. Phosphorylation of Atg32 and Atg36 is crucial for induction of mitophagy and pexophagy, respectively; the mitophagy receptor Atg32 is phosphorylated by Hog1 [37]: the pexophagy receptor Atg36 is phosphorylated by Hrr25 [38]. While it is unknown whether Hrr25 activity is changed under ER-stressed conditions, Hog1 activity is upregulated during ER stress response [7, 10]. Based on the previous findings and our results presented here, ER stress may be capable of activation of pexophagy and especially mitophagy. However, it remains to be elucidated how expression of *ATG32* and *ATG36* is induced by ER stress and whether mitophagy and pexophagy are physiologically important for ER stress response.

### The relative functional significance of Atg39 and Atg40 in ER-phagy

Here, we compared the relative contributions of Atg39 and Atg40 to ER stress- and nitrogen starvation-induced ER-phagy using several GFP-tagged proteins such as Sec63 and Rtn1. Sec63 and Rtn1 reside distinct regions: Sec63 localizes to ER sheets; Rtn1 localizes to tubules

and sheet edges of the ER. On ER stress-induced degradation of Sec63 and Rtn1, *atg39* mutation had greater inhibitory effects than *atg40* mutation (Fig 1B and S2I Fig). These results suggest that Atg39 plays a major role in ER stress-induced ER-phagy. On the other hand, *atg40* single mutation effectively inhibited nitrogen starvation-induced degradation of Sec63 and Rtn1, whereas *atg39* single mutation did not affect it (Fig 1C and S2L Fig). These results suggest that Atg40 plays a major role in nitrogen starvation-induced ER-phagy. Besides, we found that autophagic degradation of Rtn1 and Yop1 was induced by ER stress even in *atg39 atg40* double mutant cells (S2B and S2E Fig), implying the existence of unidentified autophagy receptor functioning in yeast ER-phagy induced by ER stress. Thus, the relative functional significance of ER-phagy receptors seems to be changed by the kinds of stimuli.

A previous report [20] showed the cellular localization of Atg39 and Atg40: Atg39 specifically localizes to the perinuclear ER; a large proportion of Atg40 localizes to the cortical and cytoplasmic ER, whereas its small proportion localizes to the perinuclear ER. Consistently, we found that Atg39 plays a major role in degradation of the perinuclear ER protein Hmg1 and the inner nuclear membrane protein Src1 induced by ER stress and nitrogen starvation (S2J, S2K, S2M and S2N Fig). We also detected that Atg39 significantly contributes to Rtn1 degradation in ER stress response (S2I Fig). Because Rtn1 colocalizes well with Atg40 but displays the localization pattern distinct from Atg39 after rapamycin treatment [20], it may be anticipated that Atg40 exclusively functions to degrade Rtn1, and further, our results seem to be unexpected. One possibility to explain this seeming discrepancy is that unlike rapamycin-treated cells, Atg39 may exist throughout the ER in ER-stressed cells. Alternatively, the ability of Atg39 to induce ER-phagy may be greatly enhanced or that of Atg40 may be kept low during ER stress response, compared with nitrogen starvation. However, the mRNA levels of *ATG39* and *ATG40* are inconsistent with their ability to induce ER-phagy; stronger induction of *ATG39* was seen during nitrogen starvation (compare Fig 3F with Fig 2A and 2E): higher expression of *ATG40* was observed after ER stress treatment (compare Fig 2B and 2F with S11 Fig). Therefore, it is suggested that their ability to induce ER-phagy may be regulated post-transcriptionally even if it occurs. Phosphorylation of Atg32 and Atg36 autophagy receptors is crucial for activation of mitophagy and pexophagy, respectively [37, 38]. Until now, it remains unknown whether ER-phagy receptors are regulated through phosphorylation. However, it is possible that phosphorylation is a critical determinant for the ability of Atg39 and Atg40 to induce ER-phagy, because multiple kinases are activated by ER stress and nitrogen starvation. Additionally, our data imply that ER stress causes Atg39 posttranslational modification except for phosphorylation (S5B Fig).

A previous study identified CCPG1 as an ER-phagy receptor in mammalian ER stress response [39]. Intriguingly, only mammalian CCPG1 and yeast Atg39 have long luminal tails among six mammalian and two yeast ER-phagy receptors. Therefore, it is plausible that their long luminal tails sense accumulation of aberrant protein within the ER and function to induce ER-phagy. Thus, it would enhance our understanding of the evolutionarily conserved principles in ER-phagy to investigate the difference of the mechanisms by which yeast ER-phagy receptors induce ER-phagy during nitrogen starvation and ER stress response.

### *ATG39* expression is regulated by the Snf1-Mig1/2 axis

The mRNA levels of the core autophagy-related genes are increased after nitrogen starvation and rapamycin treatment. Expression of the core autophagy-related genes is controlled at both the transcriptional and post-transcriptional levels [40–43]. Like the core autophagy-related genes, expression of autophagy receptors is increased under conditions that induce selective autophagy [17, 19, 20]. However, little is known about how it is regulated, although there are a

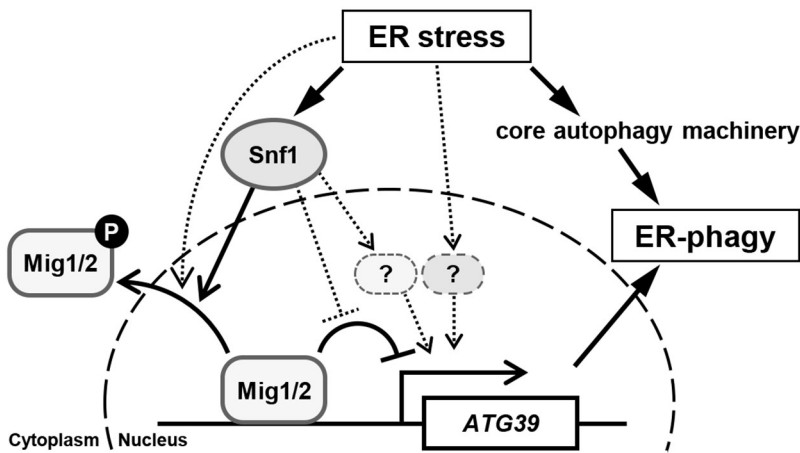

**Fig 7. Proposed model for Snf1 and Mig1/2 in ER stress-induced ER-phagy.**

few previous reports showing that *ATG40* expression is regulated by the Pho23-Rpd3L histone deacetylase complex and that *ATG32* expression is regulated by TORC1 and the Rpd3L histone deacetylase complex [22, 36]. In this study, we identified Snf1 AMPK and two closely related transcriptional repressors, Mig1 and Mig2, as a regulator of *ATG39* expression. Previous reports have revealed that Snf1 phosphorylates Mig1 and Mig2, thereby promoting their export from the nucleus [24, 25, 31]. Furthermore, we have previously shown that Snf1 is activated by ER stress [10]. Consistent with the previous findings, we revealed here that nuclear localization of Mig1 and Mig2 is reduced upon exposure to ER stress, and this reduction is suppressed by *snf1* mutation. We also found that three putative binding motifs for Mig1 and Mig2 exist in the *ATG39* promoter, and their mutations upregulate *ATG39* promoter activity, similar to *mig1 mig2* mutations. Altogether, our results suggest that *ATG39* promoter activity is repressed by Mig1 and Mig2 under unstressed conditions and derepressed through nuclear export of Mig1 and Mig2 mediated by Snf1 under ER-stressed conditions (Fig 7). Mammalian AMPK is involved in transcriptional activation of autophagy-related genes by inhibiting the transcriptional repressor BRD4 [44]. Thus, Snf1/AMPK have evolutionarily conserved roles in derepression of genes involved in autophagy.

However, our results also suggest that Mig1 and Mig2 may be regulated by unidentified molecules in addition to Snf1, since the cellular localization of Mig1 and Mig2 was not stringently consistent with their phosphorylation levels (compare Fig 5C and 5D with Fig 5A and 5B): for example, nuclear-localized Mig1 was reduced by ER stress in *snf1* mutant cells, although *snf1* mutation abolished Mig1 phosphorylation. Nuclear export of Mig1 in a manner independent of Snf1 has been predicted previously [45]. Therefore, such mechanism may operate in ER stress response. Additionally, *ATG39* expression levels did not exactly reflect the cellular localization of Mig1 and Mig2. It has been demonstrated that Mig1 physically interacts with the Cyc8-Tup1 corepressor complex to repress transcription of the target genes, and their interaction is abolished by Snf1-mediated Mig1 phosphorylation [46]. Based on this model, Mig1 may be less active in ER-stressed wild-type cells than expected from its remaining nuclear accumulation.

It is likely that an additional molecule functions downstream of Snf1 to regulate *ATG39* expression based on the following observations: first, *snf1* mutation still downregulated *ATG39* induction caused by ER stress in the *mig1 mig2* double mutant cells (Fig 4C); second, Snf1 hyperactivation caused by *reg1* mutation led to a greater increase in *ATG39* expression

than loss of Mig1 and Mig2 (compare Fig 3C and S6A Fig with S6C Fig, for example); third, *ATG39* upregulation caused by *reg1* mutation was only partly suppressed by expression a phospho-defective form of Mig1 in cells lacking Mig2 (S8F Fig). Many transcription factors including Mig1 and Mig2 are regulated by Snf1, and their target genes are partly overlapping [24, 25]. Therefore, further identification of transcription factors acting between Snf1 and the *ATG39* gene may requires combined disruption of known Snf1 targets. Furthermore, the observation that *ATG39* expression was significantly upregulated in the *snf1* mutant cells suggest the existence of a Snf1-independent mechanism inducing *ATG39* expression (Fig 3A). Intriguingly, we found that nitrogen starvation-induced *ATG39* expression is unaffected by *snf1* mutation, suggesting that a Snf1-independent mechanism principally regulates *ATG39* expression during nitrogen starvation. The difference of Snf1 dependency in *ATG39* induction between ER stress and nitrogen starvation may not be explained by Snf1 activity, since Snf1 was activated by nitrogen starvation. In mammal, expression of CCPG1 acting as an ER-phagy receptor during ER stress response is upregulated by ER stress, but the underlying mechanisms remains unclear [39]. Taken together, elucidation of the mechanisms for inducing *ATG39* expression during ER stress and nitrogen starvation will be valuable to understand how environmental stimuli induce expression of ER-phagy receptors.

## Materials and methods

### Plasmids

To make the $P_{ATG39}$-*GFP* construct, a 990-bp genomic fragment containing the 5' upstream sequences of the *ATG39* gene was amplified from the *Saccharomyces cerevisiae* W303 derivative by PCR with the following primers: 5'-CTCTAGAGGATCCCCGGGAAAAACTGTGCTCCTAGCAG-3' and 5'-TAACCCGGGGGATCCGTGACATTTTAGGTCCGACAACTCG-3'. A DNA fragment encoding GFP followed by the *ADH1* terminator ($T_{ADH1}$) was amplified from the pFA6a-GFP-HIS3MX6 vector by PCR with the following primers: 5'-CGGATCCCCGGGTTAATTAAC-3' and 5'-TCGAGCTCGGTACCCGGGAGATCTATATTACCCTGTTATCC-3'. The amplified 5' upstream sequences of the *ATG39* gene, together with the *GFP*-$T_{ADH1}$ DNA fragment, were fused to the YCplac33 vector by In-Fusion cloning kits (Takara), yielding the YCplac33-$P_{ATG39}$-GFP plasmid. Mutations of MBMs in the *ATG39* promoter were generated by oligonucleotide-directed PCR using the YCplac33-$P_{ATG39}$-GFP plasmid as a template. The primers used to mutate MBMs are: 5'-TTCGGCGATGTCCTGCGACGATTTC-3' and 5'-CAGGACATCGCCGAACGGCGAAAGC-3' for MBM1 mutation; 5'-TCGGTGTGCCATCGGTCCATCTATCC-3' and 5'-CCGATGGCACACCGATGTAGGGGGTC-3' for MBM2 and MBM3 mutations. The $P_{ATG39}$-*ATG39* construct was generated as follows. A 990-bp genomic fragment containing the *ATG39* promoter was amplified from the YCplac33-$P_{ATG39}$-GFP plasmid by PCR with the following primers: 5'-CTCTAGAGGATCCCCGGGAAAAACTGTGCTCCTAGCAG-3' and 5'-GTCTTCTTCTGACATTTTAGGTCCGACAACTCGAC-3'. The coding region of the *ATG39* gene together with the 3' downstream sequence was amplified from the *Saccharomyces cerevisiae* W303 derivative by PCR with the following primers: 5'-ATGTCAGAAGAAGACGATCATTGG-3' and 5'-TCGAGCTCGGTACCCGGGGTTTTCCTCCAGGTCTCTGTC-3'. The amplified $P_{ATG39}$ and *ATG39* DNA fragments were fused with the YCplac33 vector by In-Fusion cloning kits (Takara), yielding the YCplac33-$P_{ATG39}$-ATG39 plasmid. To generate the integrations, the inserts in the YCplac33 plasmids were subcloned into the pRS306 vector.

The *MIG1* gene was amplified from the *Saccharomyces cerevisiae* W303 derivative by PCR with the following primers: 5'- CTCTAGAGGATCCCCGGGCGGCTTGTTTAGTTGCTAGC-3' and 5'- TCGAGCTCGGTACCCGGGCTTAACTAGAGCAACCGATGC-3'. The

amplified *MIG1* DNA fragment was inserted into the YCplac33 vector by In-Fusion cloning kits (Takara), yielding the YCplac33-MIG1 plasmid. Mutations in *MIG1* were generated by oligonucleotide-directed PCR using the YCplac33-MIG1 plasmid as a template. To make the *mig1(SATA)-Myc*, *mig1(SATA)-GFP* and *mig1(SETE)-GFP* constructs, DNA fragments containing the 5' upstream and coding sequences of *MIG1* were amplified from the YCplac33-mig1(SATA) and YCplac33-mig1(SETE) plasmids by PCR with the following primers: 5'-GGTCGACGGATCCCCGGGCTTGTTTAGTTGCTAGC-3' and 5'- ACCGTTAATTAACC CGGGACTAGAGCAACCGATGC-3'. The amplified 5' upstream and coding sequences of *MIG1* were fused to the pFA6a-GFP-kanMX6 and pFA6a-13Myc-kanMX6 vectors by In-Fusion cloning kits (Takara), yielding the pFA6a-mig1(SATA)-13Myc-kanMX6, pFA6a-mig1 (SATA)-GFP-kanMX6 and pFA6a-mig1(SETE)-GFP-kanMX6 plasmids. Plasmids used in this study are described in S1 Table.

## Strains

Standard procedures were followed for yeast manipulations [47]. SD(−N) medium (0.17% (w/v) yeast nitrogen base without amino acids and ammonium sulfate and 2% (w/v) glucose) was used to induce nitrogen starvation. Yeast strains harboring the complete gene deletions (*ATG1*, *ATG11*, *ATG32*, *ATG36*, *ATG39*, *ATG40*, *MIG1* and *MIG2*) and carboxyl-terminally tagged genes (*SEC63-GFP*, *RTN1-GFP*, *YOP1-GFP*, *HMG1-GFP*, *SRC1-GFP*, *IDH1-GFP*, *PEX11-GFP*, *PGK1-GFP*, *MIG1-GFP*, *MIG2-GFP*, *MIG1-Myc*, *MIG2-Myc* and *ATG39-Myc*) were generated by a PCR-based method as described previously [48]. Primer sets were designed such that 46 bases at the 5' end of primers were complementary to those at the corresponding region of the target gene, and 20 bases at their 3' end were complementary to the pFA6a sequence, 5'-TGCAGTACTCTGCGGGTGTATACAG-3' or 5'- ATTTGACTGTATTA CCAATGTCAGC-3'. All strains produced by a PCR-based method were verified by colony PCR amplification to confirm that replacement had occurred at the expected locus. Strains carrying the $P_{ATG39}$-*GFP* and $P_{ATG39}$-*ATG39* integrations were constructed by integrating the linearized pRS306-$P_{ATG39}$-GFP and pRS306-$P_{ATG39}$-ATG39 plasmids, respectively. Strains carrying *mig1(SATA)-Myc*, *mig1(SATA)-GFP* and *mig1(SETE)-GFP* were generated as follows. The *MIG1* 5' upstream and coding sequences with Myc- or GFP-tag and *kanMX6* were amplified by PCR from the pFA6a-mig1(SATA)-13Myc-kanMX6, pFA6a-mig1(SATA)-GFP-kanMX6 and pFA6a-mig1(SETE)-GFP-kanMX6 plasmids with the following primers: 5'-TAGCATACTTGTTCGAGCTCTTGAG-3' and 5'-CTATTGTCTTTTGATTTATCTGCACC GCCAAAAACTTGTCAGCGTAGAATTCGAGCTCGTTTAAAC-3'. The amplified DNA fragments were introduced into the *mig1Δ::HIS3MX6* haploid cells, yielding *mig1(SATA)-Myc*, *mig1(SATA)-GFP* and *mig1(SETE)-GFP* strains. Strains used in this study are listed in S2 Table.

## RNA isolation and RT−PCR

Preparation of total RNA and generation of cDNA were performed as described previously [10]. The cDNAs were quantitated by a quantitative real-time RT-PCR (qRT-PCR) method using 7500 fast and QuantStudio 5 real-time RT-PCR systems (Applied Biosystems) with SYBR Premix Ex Taq (Takara), and levels of gene expression were normalized to *ACT1* expression. Primers used to analyze the mRNA levels are described in S3 Table.

## Protein extraction, western blot analysis and antibodies

Preparation of protein extracts and Western blot analysis were performed as described previously [10]. WIDE RANGE Gel Preparation Buffer(4x) for PAGE (Nakalai) was used to detect

degradation of GFP-tagged proteins. Anti-GFP monoclonal antibody JL-8 (Clontech), anti-GFP antibody from mouse IgG1κ (clones 7.1 and 13.1) (Roche), anti-phospho-AMPKα mono-clonal antibody 40H9 (Cell Signaling), anti-Snf1 polyclonal antibody yk-16 (Santa Cruz), anti-Myc monoclonal antibody 9E10 (Santa Cruz) and anti-Mcm2 polyclonal antibody N-19 (Santa Cruz) were used. Detection was carried out by using a LAS-4000 (Fuji Film) with Immobilon Western (Merck Millipore) or the Odyssey Imaging Systems (LI-COR Biosci-ences). Signal intensities were quantified by the Odyssey Imaging Systems, and statistical analysis was performed with Excel (Microsoft).

## Microscopy

To visualize GFP-tagged Mig1 and Mig2 in living cells, cells were grown at 25 ˚C until expo-nential phase and treated with 3 μg/ml tunicamycin (TM) for 3 hr. Cells were then harvested, suspended in SD medium, and observed immediately using a Keyence BZ-X700 microscope (Keyence Corporation, Japan) with a PlanAproλ 100× NA 1.45 oil objective lens (Nikon). Fluorescence intensities were quantified using Hybrid Cell Count BZ-H2C software (Keyence Corporation, Japan).

## Stress sensitivity

Assays for tunicamycin toxicity were carried out as follows. Cells were grown to exponential phase, and cultures were adjusted to an optical density of 0.5. Cell cultures were then serially diluted 5-fold, spotted onto normal plates or plates containing the indicated concentrations of tunicamycin, followed by incubation at 25˚C for 3 days (for plates lacking tunicamycin) or more than 5 days (for plates containing tunicamycin).

## Supporting information

**S1 Table. Plasmids used in this study.**
(XLSX)

**S2 Table. Strains used in this study.**
(XLSX)

**S3 Table. Primers used in this study.**
(XLSX)

**S1 Fig. Sec63-GFP degradation after ER stress treatment.** (A) Wild-type strains harboring GFP-tagged *SEC63* were grown at 25 ˚C until exponential phase and treated with 3 μg/ml tuni-camycin (TM) for the indicated time. Extracts prepared from each cell were immunoblotted with anti-GFP antibodies. (B) Wild-type (WT) and indicated mutant strains harboring GFP-tagged *SEC63* were grown at 25 ˚C until exponential phase and treated with 6 mM dithiothrei-tol (DTT) for 18 hr. Extracts prepared from each cell were immunoblotted with anti-GFP anti-bodies.
(TIF)

**S2 Fig. Degradation of ER-localized proteins after ER stress treatment and nitrogen star-vation.** (A) Cellular localization of Sec63, Hmg1, Src1, Rtn1 and Yop1. Wild-type harboring GFP-tagged *SEC63*, *HMG1*, *SRC1*, *RTN1* or *YOP1* were grown at 25 ˚C until exponential phase and subjected to microscopy. (B-E) Degradation of Rtn1-GFP (B), Hmg1-GFP (C), Src1-GFP (D), and Yop1-GFP (E) after ER stress treatment. Wild-type (WT) and indicated mutant strains harboring GFP-tagged *RTN1*, *HMG1*, *SRC1*, or *YOP1* were grown at 25 ˚C until exponential phase and treated with 3 μg/ml tunicamycin (TM) for 18 hr. Extracts

prepared from each cell were immunoblotted with anti-GFP antibodies. The intensities of free GFP were measured and normalized to the intact GFP-tagged protein level. The values are plotted as the fold change from wild-type cells. The data show mean ± SEM (n > 3). *$P < 0.05$ and **$P < 0.01$ as determined by Student's $t$-test. (F-H) Degradation of Rtn1-GFP (F), Hmg1-GFP (G), and Src1-GFP (H) after nitrogen starvation. Wild-type (WT) and indicated mutant strains harboring GFP-tagged *RTN1*, *HMG1*, or *SRC1* were grown at 25 ˚C until exponential phase and then incubated under nitrogen-starved conditions for 18 hr. Extracts prepared from each cell were immunoblotted with anti-GFP antibodies. The intensities of free GFP were measured and normalized to the intact GFP-tagged protein level. The values are plotted as the fold change from wild-type cells. The data show mean ± SEM (n > 3). **$P < 0.01$ as determined by Student's $t$-test. (I-K) Degradation of Rtn1-GFP (I), Hmg1-GFP (J), and Src1-GFP (K) after ER stress treatment. Wild-type (WT) and indicated mutant strains harboring GFP-tagged *RTN1*, *HMG1*, or *SRC1* were grown at 25 ˚C until exponential phase and treated with 3 µg/ml tunicamycin (TM) for 18 hr. Extracts prepared from each cell were immunoblotted with anti-GFP antibodies. The intensities of free GFP were measured and normalized to the intact GFP-tagged protein level. The values are plotted as the fold change from wild-type cells. The data show mean ± SEM (n > 3). **$P < 0.01$ as determined by Student's $t$-test. NS, not statistically significant ($P > 0.05$). (L-N) Degradation of Rtn1-GFP (L), Hmg1-GFP (M), and Src1-GFP (N) after nitrogen starvation. Wild-type (WT) and indicated mutant strains harboring GFP-tagged *RTN1*, *HMG1*, or *SRC1* were grown at 25 ˚C until exponential phase and then incubated under nitrogen-starved conditions for 18 hr. Extracts prepared from each cell were immunoblotted with anti-GFP antibodies. The intensities of free GFP were measured and normalized to the intact GFP-tagged protein level. The values are plotted as the fold change from wild-type cells. The data show mean ± SEM (n > 3). **$P < 0.01$ as determined by Student's $t$-test.
(TIF)

**S3 Fig. The *ATG39* mRNA levels after ER stress treatment.** (A-C) Wild-type (WT) and indicated mutant strains were grown at 25 ˚C until exponential phase and treated with 6 mM dithiothreitol (DTT) for the indicated time. The *ATG39* mRNA levels were quantified by qRT-PCR analysis, and relative mRNA levels were calculated using *ACT1* mRNA. The values are plotted as the fold change from wild-type cells at the time of DTT addition. The data show mean ± SEM (n > 3). **$P < 0.01$ as determined by Student's $t$-test.
(TIF)

**S4 Fig. Effects of *snf1* mutation on the mRNA levels of the genes encoding a selective autophagy receptor.** (A-D) Wild-type (WT) and *snf1* mutant strains were grown at 25 ˚C until exponential phase and treated with 3 µg/ml tunicamycin (TM) for the indicated time. The mRNA levels were quantified by qRT-PCR analysis, and relative mRNA levels were calculated using *ACT1* mRNA. The values are plotted as the fold change from wild-type cells at the time of TM addition. The data show mean ± SEM (n > 3). *$P < 0.05$ and **$P < 0.01$ as determined by Student's $t$-test. NS, not statistically significant ($P > 0.05$).
(TIF)

**S5 Fig. The Atg39 protein after ER stress treatment.** (A) The Atg39 protein level after ER stress treatment. Wild-type strains harboring non-tagged or Myc-tagged *ATG39* were grown at 25 ˚C until exponential phase and treated with 6 mM dithiothreitol (DTT) for the indicated time. (B) Effects of the phosphatase treatment on Atg39. Wild-type strains harboring Myc-tagged *ATG39* were grown at 25 ˚C until exponential phase and treated with 6 mM dithiothreitol for 6 hr. Extracts were treated with or without calf intestinal alkaline phosphatase and

subjected to immunoblot with anti-Myc antibodies.
(TIF)

**S6 Fig. The *ATG39* mRNA levels in mutants of known Snf1 targets.** (A, B) The *ATG39* mRNA levels in mutants of known Snf1 targets. Wild-type (WT) and indicated mutant strains were grown at 25 ˚C until exponential phase. The *ATG39* mRNA levels were quantified by qRT-PCR analysis, and relative mRNA levels were calculated using *ACT1* mRNA. The values are plotted as the fold change from wild-type cells. The data show mean ± SEM (n = 3). (C) The *ATG39* mRNA levels in unstressed *mig1 mig2* mutant. Wild-type (WT) and indicated mutant strains were grown at 25 ˚C until exponential phase. The *ATG39* mRNA levels were quantified by qRT-PCR analysis, and relative mRNA levels were calculated using *ACT1* mRNA. The values are plotted as the fold change from wild-type cells. The data show mean ± SEM (n = 4). $^{**}P < 0.01$ as determined by Student's *t*-test.
(TIF)

**S7 Fig. Mig1/2 repress *ATG39* expression.** (A) The *ATG39* mRNA levels in ER-stressed *mig1 mig2* mutant. Wild-type (WT) and *mig1 mig2* mutant strains were grown at 25 ˚C until exponential phase and treated with 3 μg/ml tunicamycin (TM) for the indicated time. The *ATG39* mRNA levels were quantified by qRT-PCR analysis, and relative mRNA levels were calculated using *ACT1* mRNA. The values are plotted as the fold change from wild-type cells at the time of TM addition. The data show mean ± SEM (n = 5). $^{**}P < 0.01$ as determined by Student's *t*-test. (B) Effects of mutations in putative Mig1/2-binding motifs on ER stress-induced *ATG39* upregulation. Wild-type (WT) cells harboring the integration which expresses GFP under the control of wild-type or mutated *ATG39* promoter were grown at 25 ˚C until exponential phase and treated with 6 mM dithiothreitol (DTT) for the indicated time. The *GFP* mRNA levels were quantified by qRT-PCR analysis, and relative mRNA levels were calculated using *ACT1* mRNA. The values are plotted as the fold change from wild-type cells harboring the integration which expresses GFP under the control of wild-type *ATG39* promoter at the time of DTT addition. The data show mean ± SEM (n = 4). NS, not statistically significant ($P > 0.05$), as determined by Student's *t*-test.
(TIF)

**S8 Fig. The electrophoretic mobility patterns, the protein levels and localization of Mig1.** (A, B) The protein levels of Mig1 (A) and Mig2 (B). Wild-type (WT) and *snf1* mutant strains harboring GFP-tagged *MIG1* (A) or *MIG2* (B) were grown at 25 ˚C until exponential phase, treated with 3 μg/ml tunicamycin (TM) for 3 hr. Extracts prepared from each cell were immunoblotted with anti-Myc antibodies. The intensities of Mig1-Myc and Mig2-Myc were measured and normalized to the Mcm2 level. The values are plotted as the fold change from wild-type cells at the time of TM addition. (C) Effects of the phosphatase treatment on Mig1. Wild-type (WT) and *snf1* mutant strains harboring Myc-tagged *MIG1* were grown at 25 ˚C until exponential phase and treated with 3 μg/ml tunicamycin (TM) for 3 hr. Extracts prepared from each cell were immunoblotted with anti-Myc antibodies. (D) Cellular localization of Mig1 in *reg1* and *reg1 snf1* mutants. Wild-type (WT) and indicated mutant strains harboring GFP-tagged *MIG1* were grown at 25 ˚C until exponential phase and subjected to microscopy. (E) Cellular localization of Mig1 mutated in putative Snf1 phosphorylation sites. Wild-type (WT) and *reg1* mutant strains harboring GFP-tagged *MIG1* were grown at 25 ˚C until exponential phase and subjected to microscopy. (F) Effects of the phospho-defective mutation of Mig1 on *ATG39* upregulation caused by *reg1* mutation. The *mig2* mutant strains harboring Myc-tagged wild-type or the phospho-defective mutant *MIG1* were grown at 25 ˚C until exponential phase. The *ATG39* mRNA levels were quantified by qRT-PCR analysis, and relative

mRNA levels were calculated using *ACT1* mRNA. The values are plotted as the fold change from *MIG1(WT) mig2* cells. The data show mean ± SEM (n = 4). **$P < 0.01$ as determined by Student's *t*-test. NS, not statistically significant ($P > 0.05$).
(TIF)

**S9 Fig. Degradation of GFP-tagged proteins in ER-stressed cells.** (A) Sec63-GFP degradation in Snf1-activated cells. Wild-type (WT) and indicated mutant strains harboring GFP-tagged *SEC63* were grown at 25 ˚C until exponential phase and treated with 3 μg/ml tunicamycin (TM) for 18 hr. Extracts prepared from each cell were immunoblotted with anti-GFP antibodies. The intensities of free GFP were measured and normalized to the Sec63-GFP level. The values are plotted as the fold change from wild-type cells. The data show mean ± SEM (n = 3). *$P < 0.05$ and **$P < 0.01$ as determined by Student's *t*-test. (B) Pgk1-GFP degradation in ER-stressed *snf1* mutant. Wild-type (WT) and indicated mutant strains harboring GFP-tagged *PGK1* were grown at 25 ˚C until exponential phase and treated with 3 μg/ml tunicamycin (TM) for 18 hr. Extracts prepared from each cell were immunoblotted with anti-GFP antibodies. The intensities of free GFP were measured and normalized to the Pgk1-GFP level. The data show mean ± SEM (n = 4). NS, not statistically significant ($P > 0.05$), as determined by Student's *t*-test. (C, D) Effects of *snf1* mutation on Sec63-GFP degradation in the *atg39* and *atg40* mutant cells. Indicated mutant strains harboring GFP-tagged *SEC63* were grown at 25 ˚C until exponential phase and treated with 3 μg/ml tunicamycin (TM) for 18 hr. Extracts prepared from each cell were immunoblotted with anti-GFP antibodies. The intensities of free GFP were measured and normalized to the Sec63-GFP level. The values are plotted as the fold change from the *atg39* mutant (C) or the *atg40* mutant (D). The data show mean ± SEM (n = 4). **$P < 0.01$ as determined by Student's *t*-test. NS, not statistically significant ($P > 0.05$). (E) Sec63-GFP degradation in *mig1 mig2* mutant. A dark, high-contrast image of Fig 6C is shown. (F) GFP-Atg8 degradation in *mig1 mig2* mutant. Wild-type (WT) and *mig1 mig2* mutant strains harboring GFP-tagged *ATG8* were grown at 25 ˚C until exponential phase and treated with or without 3 μg/ml tunicamycin (TM) for 6 hr. Extracts prepared from each cell were immunoblotted with anti-GFP antibodies. (G) Hmg1-GFP degradation in *mig1 mig2* mutant. Wild-type (WT) and indicated mutant strains harboring GFP-tagged *HMG1* were grown at 25 ˚C until exponential phase and treated with 3 μg/ml tunicamycin (TM) for 18 hr. Extracts prepared from each cell were immunoblotted with anti-GFP antibodies. The intensities of free GFP were measured and normalized to the Hmg1-GFP level. The values are plotted as the fold change from wild-type cells. The data show mean ± SEM (n = 4). **$P < 0.01$ as determined by Student's *t*-test. NS, not statistically significant ($P > 0.05$). A dark, high-contrast image is also shown. (H) Effects of the phospho-defective mutation of Mig1 on Hmg1-GFP degradation. The *mig2* mutant strains harboring GFP-tagged *HMG1* and Myc-tagged wild-type or the phospho-defective mutant *MIG1* were grown at 25 ˚C until exponential phase and treated with 3 μg/ml tunicamycin (TM) for 18 hr. Extracts prepared from each cell were immunoblotted with anti-GFP antibodies. The intensities of free GFP were measured and normalized to the Hmg1-GFP level. The values are plotted as the fold change from *MIG1 (WT) mig2* cells. The data show mean ± SEM (n = 4). **$P < 0.01$ as determined by Student's *t*-test. (I) Hmg1-GFP degradation in cells expressing *ATG39* under the control of mutated *ATG39* promoter. The *atg39* mutants harboring GFP-tagged *SEC63* and the integration which expresses *ATG39* under the control of wild-type or mutated *ATG39* promoter were grown at 25 ˚C until exponential phase and treated with 3 μg/ml tunicamycin (TM) for 18 hr. Extracts prepared from each cell were immunoblotted with anti-GFP antibodies. The intensities of free GFP were measured and normalized to the Sec63-GFP level. The values are plotted as the fold change from cells harboring the wild-type *ATG39* integration. The data show mean ± SEM

(n = 4). $^{*}P < 0.05$ as determined by Student's $t$-test.
(TIF)

**S10 Fig. Effects of *atg39* and *atg40* mutations on ER stress sensitivity.** (A, B) Wild-type (WT) and indicated mutant strains were spotted onto YPD medium lacking or containing 0.5 or 1.5 μg/ml tunicamycin (TM) and incubated at 25 ˚C.
(TIF)

**S11 Fig. The *ATG40* mRNA levels after nitrogen starvation.** Wild-type strains were grown at 25 ˚C until exponential phase and then incubated under nitrogen starvation conditions for the indicated time. The mRNA levels were quantified by qRT-PCR analysis, and relative mRNA levels were calculated using *ACT1* mRNA. The values are plotted as the fold change from untreated cells. The data show mean ± SEM (n = 3). $^{**}P < 0.01$ as determined by Student's $t$-test.
(TIF)

## Acknowledgments

We thank the members of our laboratory for their helpful suggestions and feedback.

## Author Contributions

**Conceptualization:** Tomoaki Mizuno.

**Formal analysis:** Tomoaki Mizuno.

**Funding acquisition:** Tomoaki Mizuno.

**Investigation:** Tomoaki Mizuno, Kei Muroi.

**Methodology:** Tomoaki Mizuno.

**Project administration:** Tomoaki Mizuno.

**Resources:** Tomoaki Mizuno, Kenji Irie.

**Supervision:** Tomoaki Mizuno.

**Validation:** Tomoaki Mizuno.

**Visualization:** Tomoaki Mizuno.

**Writing – original draft:** Tomoaki Mizuno.

**Writing – review & editing:** Tomoaki Mizuno, Kenji Irie.

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
