## [Decision Letter · Decision Letter 0]

30 Apr 2020

Dear Dr Mizuno,

Thank you very much for submitting your Research Article entitled 'Snf1 AMPK positively regulates ER-phagy via expression control of Atg39 autophagy receptor in yeast ER stress response' to PLOS Genetics. Your manuscript was fully evaluated at the editorial level and by independent peer reviewers. The reviewers appreciated the attention to an important topic but identified some aspects of the manuscript that should be improved.

All three reviewers stress the fact that Sec63 may not be the best readout for this analysis. As suggested, Rtn1, or some other resident ER protein, should be used in addition. Moreover, both the second and third reviewer noted that the epistasis of Mig1/2 phosphomutants with snf1∆ would better explore your model. In addition, reviewer 3 noted that determining whether deletion of SNF1 also affects upregulation of ATG32/36 will allow the reader to understand better whether SNF1 functions uniquely in ER-phagy. In addition to these experiments, the missing reference noted by the first reviewer and other aspects of the text need to be amended. We therefore ask you to modify the manuscript according to these review recommendations before we can consider your manuscript for acceptance.

[LINK]

Yours sincerely,

David P. Toczyski

Associate Editor

PLOS Genetics

Gregory P. Copenhaver

Editor-in-Chief

PLOS Genetics

Reviewer's Responses to Questions

**Comments to the Authors:**

Reviewer #1: Review-PGENETICS-D-20-00376

In this manuscript the authors claim to have shown for the first time that ER stress regulates two budding yeast ER-phagy receptors, Atg39 and Atg40. The authors show that the expression level of both receptors increases in response to ER stress. Focusing on Atg39 regulation, they go on to show that the Snf1 kinase positively regulates Atg39, while Mig1 and Mig 2 negatively regulate Atg39. The authors argue that Snf1 promotes ER-phagy by negatively regulating Mig1 and Mig2. There are many issues that need to be addressed in this manuscript.

1) It has already been reported that Atg40 responds to ER stress (Cui et al. 2019 365, 53-60 Science). The Abstract, Introduction and Discussion needs to be rewritten to correct inaccurate statements regarding this point.

2) The authors use one ER protein, Sec63, to do all their analysis. This is not the best marker to use for Atg39 as it is the nucleophagy receptor. Mochida et al. 2015 Nature showed that Atg39 primarily degrades the nucleus. Their analysis of Atg39 was done with proteins that mark the nuclear envelope (Hmg1), the inner nuclear membrane (Src1) and the nucleolus (Nop1). These markers do no label the cortical ER. A cortical ER marker, Rtn1, should also be analyzed as a negative control.

3) The authors claim that Sec63-GFP degradation induced by tunicamycin, does not depend on Atg40. The data in Figure 1B, however, shows that Atg39 and Atg40 act synergistically to degrade Sec63.

4) The authors claim that Atg39 does not play a major role during nitrogen starvation is based on the finding that they do not see degradation of Sec63-GFP (Figure 1C). They need to look at other markers (see comment # 2) before drawing this conclusion.

5) I don’t understand why tunicamycin, a drug that blocks N-linked glycosylation, induces pexophagy, mitophagy and nucleophagy (Atg39).

6) The authors state that rapamycin is “a compound that makes cells nitrogen-starved”.

This is a misleading statement. Rapamycin is a drug that mimics starvation. Rapamycin and nitrogen starvation do not always stimulate the same autophagy pathways (i.e. the requirements for rapamycin induced and nitrogen starvation induced autophagy are not always the same).

7) The data on Snf1 and Mig1 and Mig2 are not always convincing.

Reviewer #2: Previous studies have shown that ER stress induces a variety of pathways for macroautophagy, as well as ER turnover by microautophagy. An independent body of work has also led to the identification of several ER-localized receptors in budding yeast (Atg39/40) and mammalian cells that enable starvation signals to selectively target the ER for (macro)autophagic destruction. Mizuno et al. provide evidence for a novel connection between prolonged ER stress and Atg39/40-mediated ER-phagy. They specifically find an ER to nucleus signaling pathways pathway that transcriptionally up-regulates Atg39 expression under ER stress. Their data support a mechanistic model in which ER stress induces activation of the yeast AMPK Snf1 to promote nuclear export of two repressors that are normally bound to the Atg39 promoter (Mig1/2). Overall, this work rests on very concrete data that were generated by well-designed experiments but a few additional experiments and some minor revisions would help polish a revised version of this manuscript for publication.

Major points:

1. A key part of their model is that Snf1 activation due to ER stress leads to the phosphorylation of Mig1/2 repressors and their nuclear export, thereby de-repressing Atg39 transcription. The supporting data for this idea, however, also reveal additional complexities that are not captured by their model. For example, in the triple (snf1 mig1 mig2) mutant (Fig 4C), ER stress still induces a handsome level of Atg39 transcriptional induction. The authors focus on the significance of the slight decrease in the maximum of this induction that is apparently Snf1-dependent (ie. relative to mig1 mig2) but to me it is more shocking how much inducibility persists that remains unaccounted for. Along the same lines, even though in mig1 mig2 there is a ~3-fold increase in basal Atg39 transcription (Fig 2B), there is also a ~20-fold Atg39 transcriptional induction due to ER stress. Given these complexities (ie. the apparent existence of undefined ER stress signaling routes to the Atg39 promoter (independent of Snf1), as well as from Snf1 to the Atg39 promoter (independent of Mig1/2), it would behoove the authors to further validate their model in the following two ways. First, define the effect of Snf1 phosphosite mutations on Mig1/2 on basal and ER-stress induced Atg39 mRNA levels. Second, use a minimal Atg39 promoter construct (e.g. crippled promoter with MBM region) to provide a transcriptional reporter that fully captures the key aspect of their model but excludes other undefined transcriptional regulation factors that complicate the picture they are describing.

2. Figure 5 provides evidence for a subcellular localization change of Mig1 and Mig2 under ER stress (TM treatment). It is not clear to me that this relatively small 2-fold change is sufficient to drive the associated Atg39 transcriptional change (presumably close to an order of magnitude fold induction by comparison to similar conditions in other experiments shown) they are reporting. The authors should consider examining Mig1/2 protein level and phosphorylation status to rule out other contributions to Mig1/2-regulated Atg39 expression beyond nuclear export.

Minor points:

1. The authors acknowledge that “It is possible that the long duration of ER stress is required for Atg39 to express enough to induce ER-phagy.” but they should show the kinetics of the key steps in their model (1. Snf1 kinase activation; Mig1/2 nuclear export; Atg39 transcriptional induction measured as mRNA level; Atg39 protein induction) by showing them together in relationship to one another. As it stands, it is hard to define the nature of the “slow step” in their model. For example, in Fig 2, the mRNA level of Atg39 already increased by ~50-fold after at 6 hrs tunicamycin treatment, but no free GFP band was detected in the Sec63-GFP processing assay (Fig 1).

2. The authors make the broad claim that Atg39 makes the dominant contribution to ER stress-induced ER-phagy. This is in contrast to the dominant role of Atg40 in ER-phagy under nitrogen starvation. Given that previous work has shown that Atg40 is enriched in highly curved ER membranes (tubules and sheet edges vs sheets) whereas Sec63 is preferentially found in ER sheets, it is conceivable that the Sec63-GFP processing assay could be a biased metric of ER-phagy. The authors could consider defining the autophagic flux of the two ER-phagy receptors themselves as a potentially more realistic representation of their contribution to ER-phagy induced by ER stress.

minor points

1. There is visible new, lower mobility Atg39-myc species in Fig 3B that appears associated with ER stress. The authors should at the very least comment in the figure legend whether this species corresponds to a form of Atg39 (vs. a cross-reacting anti-Myc species also present in Atg39 control lacking Myc tag).

2. It would be helpful for completion to include the changes of Atg40 mRNA level in snf1∆ cells under TM treatment in supplementary figure, if these data have been already collected.

3. Actual experimental evidence of direct binding of Mig1/2 to the Atg39 promoter would help provide further support for their model. Could the authors comment on whether this is the case based on mining of published large-scale ChIP-seq data sets.

4. Please comment on the visible band shift to higher molecular weight of Sec63-GFP after 18 hr TM treatment (e.g. Fig 6D, 6E).

Text:

Line 272-273 “Taken together, these results suggest that ATG39 expression is inhibited by Mig1 and Mig2, and this inhibition is cancelled by Snf1.”

I get the meaning but it sounds clunky in English. I would consider saying “this inhibition is itself negatively regulated/controlled by Snf1.”

Line 321-323 “Thus, it is unlikely that Snf1 is needed for activation of nonselective autophagy in response to ER stress.”

I would consider saying “Thus, Snf1 appears to be dispensable of the activation of nonselective autophagy in response to ER stress.”

Line 389-390 “Our observations suggest that Atg39 is more and less important than Atg40 for ER stress- and nitrogen starvation-induced ER-phagy, respectively.”

I would say “Our observations suggest that Atg39 plays a major role in ER stress induced ER-phagy, whereas Atg40 is more important for enabling ER-phagy under nitrogen starvation.”

Reviewer #3: A great deal of recent ER-phagy research in yeast and mammalian systems focus on identifying the ER receptors (eg. Atg39 and Atg40 in yeast) that recruit autophagic components to the ER surface. However, little is known about how these different autophagy receptors are being regulated at basal versus stressed situations. In this study, Mizuno et al. focused on the upstream regulation of Atg39 and found that ER stress induces Snf1-mediated phosphorylation of the transcriptional repressors, Mig1/2. This results in Mig1/2 nuclear export and subsequently leads to the transcriptional upregulation of Atg39. This study is important as it sheds light on additional regulatory mechanisms that dictates ER-phagy activation. Overall, this study is very well executed with detailed investigation and convincing data to support their argument. While I am generally in favour of publication, I feel there are a few points that should be addressed.

Major comments

1. The major readout for ER-phagy throughout the paper is Sec63-GFP cleavage. This runs into the risk of potentially studying only the effect of Snf1/Mig1/2 regulation on Sec63 degradation rather than overall ER degradation. The authors should show the degradation of other ER resident proteins. This does not need to be done for all ER-phagy assays but the authors should at least be able to demonstrate that the degradation of another ER protein follows the same trend as Sec63 under ER stress vs nitrogen starvation with Atg39/40 mutations (similar experimental set up as Fig 1B and C).

2. In Figure 6, the authors used a PGK-GFP cleavage assay to show that Snf1 KO does not affect bulk autophagy. However, the authors showed that ER stress also induces mitophagy and pexophagy. It seems important to investigate whether KO of Snf1 also affects upregulation of Atg32 and Atg36. This will shed light on whether Snf1 is a specific ER-phagy inducer or a global organelle-autophagy inducer during ER stress.

3. In Fig 5A, the fluorescence intensity ratio (N/C) data for Mig1 is very subtle, especially if compared to control versus TM treatment of both WT and Snf1KO. Is it possible that Snf1 regulates Mig1 in a different fashion than Mig2 (i.e. apart from nuclear export)?

4. The authors draw a great deal of conclusions from previous literature showing that phosphorylation of Mig1&2 causes nuclear export. But the link between Mig1/2 phosphorylation and ER-phagy is not formally tested in this paper. This hypothesis should be checked, since it’s a major part of their model. One suggestion would be to generate phosphomimetic Mig1 and Mig2 mutants in Snf1 KO cells to see if Mig1&2 are driven out of the nucleus and trigger ER-phagy even in the absence of Snf1 KO.

5. The authors’ data very nicely shows the strong effect of Snf1 and Mig1/2 on transcription of Atg39. However, Snf1 and Mig1/2 have only a minor effect on ER-phagy. This is apparent in Fig 6A/D where Snf1 KO showed a statistically significant but relatively small inhibition of ER-phagy. So Snf1 and Mig1/2 regulate the ER-phagy receptors, but why does this not translate into marked effects on ER-phagy? The authors should try to delve deeper into how the transcriptional regulation of Atg39 correlates with ER-phagy.

6. In Fig 4C, the data do not quite fit with the model they proposed. If the sole function of Snf1 during ER-phagy is to remove Mig1 and Mig2 from the Atg39 promoter, triple KO of Snf1/Mig1/Mig2 should phenocopy Mig1/Mig2 double KO. The fact that the triple mutant showed less ATG39 activation compared to WT or Mig1+2 mutant suggests that Snf1 also upregulates ATG39 via an additional mechanism beyond Mig1 and Mig2. The authors did point this out this during discussion but I think this should also be noted in the result section and further expanded in the discussion on what other potential mechanism that could be. And this additional function of Snf1 should also be reflected in the schematic of Fig 7.

Minor comments

In general, I feel the authors need to explain their findings in more detail in the discussion. I also felt this paper delves deep into ER-phagy in yeast, but there is almost no connection made to the burst of recent papers on mammalian ER-phagy. Some sections which I recommend further discussion are as follows:

1. The authors showed in Fig.2 that ER stress (TM and DTT) causes transcriptional up regulation of all ATG receptors including ER, mitochondria and peroxisomes. Does this mean that ER stress acts as a general autophagy inducer? While this is not the main focus of the paper, this is an unexpected finding and the authors should address this observation in the discussion.

2. There is a distinct difference in the abundance of full length Sec63-GFP band between TM versus nitrogen starvation (compare Fig 1B and C). The authors should address what this means in the discussion.

3. Since double KO of Mig1 and Mig2 resulted in upregulation of Atg39 even at basal state (e.g. Fig 4A), one might expect that ER-phagy should also be enhanced at basal state but the authors did not observe this in Fig 6D. The authors should explain/discuss this.

4. In line 221-222 of the main text - The sentence ‘…rapamycin, a compound that makes cells nitrogen-starved’ is misleading since rapamycin does not cause nitrogen starvation but instead it mimics the downstream signal activation during nitrogen starvation.

5. The authors should also discuss whether there are known human homologs for Mig1 and Mig2 with conserved functions during ER-phagy.

6. There are many new publications in the past few years reporting on new mammalian ER-phagy receptors and post-translational modifications (e.g. phosphorylation, ubiquitylation and UFMylation) that form additional regulatory mechanism during ER-phagy and the authors should discuss how their discovery contributes to our existing knowledge on ER-phagy regulation.

**Have all data underlying the figures and results presented in the manuscript been provided?**

Reviewer #1: Yes

Reviewer #2: Yes

Reviewer #3: Yes

PLOS authors have the option to publish the peer review history of their article (what does this mean?). If published, this will include your full peer review and any attached files.

Reviewer #1: No

Reviewer #2: No

Reviewer #3: No

---

## [Decision Letter · Decision Letter 1]

14 Aug 2020

Dear Dr Mizuno,

We are pleased to inform you that your manuscript entitled "Snf1 AMPK positively regulates ER-phagy via expression control of Atg39 autophagy receptor in yeast ER stress response" has been editorially accepted for publication in PLOS Genetics. Congratulations!

Please note that Reviewer #2 has some remaining suggestions (see below).  We believe these are thoughtful and would strengthen the manuscript, but we are not making our decision contingent on addressing them.  Nonetheless, you should feel free (if not encouraged) to consider them as you prepare your final draft for the production team (the editorial team will not need to re-evaluate).

Yours sincerely,

David P. Toczyski

Associate Editor

PLOS Genetics

Gregory P. Copenhaver

Editor-in-Chief

PLOS Genetics

Comments from the reviewers (if applicable):

Reviewer's Responses to Questions

**Comments to the Authors:**

Reviewer #2: The authors have addressed my main concerns with new data, as well as by revising their text. The new manuscript is substantially improved and should be ready for publication after addressing (at the editorial level) the minor critiques/comments below:

1. I would use more caution in interpreting the “no effect” of phosphatase treatment shown in Fig S5B. These data are simply not convincing enough (ie. neither quantitated nor controlled with a phosphatase inhibitor) to formally exclude the possibility of phosphorylation as the visible Atg39 post-translational modification.

2. re: Atg39/40 autophagic flux data shown to reviewers only. Is there a good reason to not include them with the same transparent discussion in the manuscript?

3. The data shown in Fig S8C (supporting 5A/B), imply the existence of distinct Mig1 phosphorylation states (basal and Tm-induced) and that they are both Snf1-dependent. The authors should explicitly mention this in their text. It would also be worth showing the SATA and SETE mutants side-by-side with the analysis in Fig S8C to determine the contribution of their mutated phosphosites on Mig1’s gel migration.

4. The new Western blotting data for Hmg1-GFP and Src1-GFP in Fig S2 are of less than ideal quality (e.g. S2K). Might be worth testing if the extent of apparent protein degradation would be remedied by treating cultured cells with 10% cold TCA etc. prior to lysate preparation.

Reviewer #3: This very thorough revision has addressed all of my concerns.

**Have all data underlying the figures and results presented in the manuscript been provided?**

Reviewer #2: Yes

Reviewer #3: Yes

PLOS authors have the option to publish the peer review history of their article (what does this mean?). If published, this will include your full peer review and any attached files.

Reviewer #2: **Yes: **Vladimir Denic

Reviewer #3: No

**Data Deposition**

http://datadryad.org/submit?journalID=pgenetics&manu=PGENETICS-D-20-00376R1

**Press Queries**

---

## [Editor Report · Acceptance letter]

23 Sep 2020

PGENETICS-D-20-00376R1 

Snf1 AMPK positively regulates ER-phagy via expression control of Atg39 autophagy receptor in yeast ER stress response 

Dear Dr Mizuno, 

We are pleased to inform you that your manuscript entitled "Snf1 AMPK positively regulates ER-phagy via expression control of Atg39 autophagy receptor in yeast ER stress response" has been formally accepted for publication in PLOS Genetics! Your manuscript is now with our production department and you will be notified of the publication date in due course.

With kind regards,

Jason Norris

PLOS Genetics

On behalf of:
